# PIG: Physics-Informed Gaussians as Adaptive Parametric Mesh Representations

**Namgyu Kang***
Department of Artificial Intelligence
Yonsei University

**Jaemin Oh***
Department of Mathematical Sciences
KAIST

**Youngjoon Hong**†
Department of Mathematical Sciences
Seoul National University

**Eunbyung Park**†
Department of Artificial Intelligence
Yonsei University

## Abstract

The numerical approximation of partial differential equations (PDEs) using neural networks has seen significant advancements through Physics-Informed Neural Networks (PINNs). Despite their straightforward optimization framework and flexibility in implementing various PDEs, PINNs often suffer from limited accuracy due to the spectral bias of Multi-Layer Perceptrons (MLPs), which struggle to effectively learn high-frequency and nonlinear components. Recently, parametric mesh representations in combination with neural networks have been investigated as a promising approach to eliminate the inductive bias of MLPs. However, they usually require high-resolution grids and a large number of collocation points to achieve high accuracy while avoiding overfitting. In addition, the fixed positions of the mesh parameters restrict their flexibility, making accurate approximation of complex PDEs challenging. To overcome these limitations, we propose Physics-Informed Gaussians (PIGs), which combine feature embeddings using Gaussian functions with a lightweight neural network. Our approach uses trainable parameters for the mean and variance of each Gaussian, allowing for dynamic adjustment of their positions and shapes during training. This adaptability enables our model to optimally approximate PDE solutions, unlike models with fixed parameter positions. Furthermore, the proposed approach maintains the same optimization framework used in PINNs, allowing us to benefit from their excellent properties. Experimental results show the competitive performance of our model across various PDEs, demonstrating its potential as a robust tool for solving complex PDEs. Our project page is available at `https://namgyukang.github.io/Physics-Informed-Gaussians/`

## 1 Introduction

Machine learning techniques have become promising tools for numerical solutions to partial differential equations (PDEs) (Raissi et al., 2017; Yu et al., 2018; Karniadakis et al., 2021; Finzi et al., 2023; Gaby et al., 2024). A notable example is the Physics-Informed Neural Network (PINN) (Raissi et al., 2019), which leverages Multi-Layer Perceptrons (MLPs) and gradient-based optimization algorithms. This approach circumvents the need for the time-intensive mesh design prevalent in numerical methods and allows us to solve both forward and inverse problems within the same optimization framework. With the increased computational power and the development of easy-to-use automatic differentiation software libraries (Abadi et al., 2015; Bradbury et al., 2018; Innes, 2018; Paszke et al., 2019), PINNs have successfully tackled a broad range of challenging PDEs (Hu et al., 2024c; Li et al., 2024; Oh et al., 2024).

Although the neural network approach shows significant promise in solving PDEs, it has several limitations. Training PINNs typically requires numerous iterations to converge (Saarinen et al., 1993;

---

*Equal contribution
†Corresponding authors

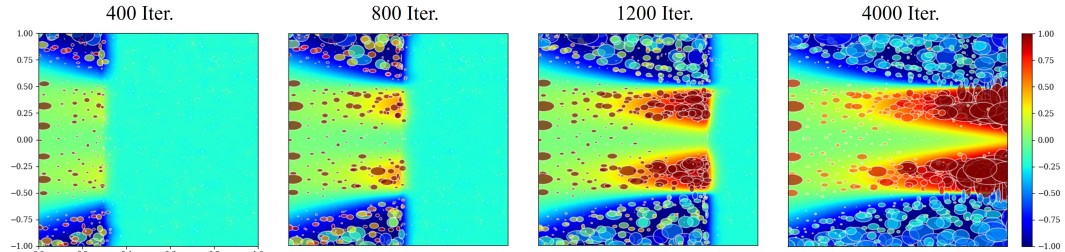

Figure 1: Training visualization of the Allen-Cahn equation (400, 800, 1200, 4000 training iterations): Each Gaussian is displayed as the ellipsoids, exhibiting different positions and shapes according to the Gaussian parameters, mean and covariance. Since we adopt a causal loss (Wang et al., 2024c), the solution is gradually approximated from $t = 0$ to $t = 1$. Note that the Gaussians are densely aligned in the locations where the solution changes abruptly.

Wang et al., 2021; De Ryck et al., 2023). Despite recent techniques aimed at reducing computational costs, multiple forward and backward passes of neural networks are still necessary to update network parameters. Furthermore, obtaining more accurate approximations demands the use of wider and deeper neural networks, which enhances their expressiveness but significantly increases computational costs (Cybenko, 1989; Baydin et al., 2018; Kidger & Lyons, 2020). In addition, inductive biases inherent in MLPs often hinder the accuracy of solution approximations. A well-known example is the spectral bias, which favors learning low-frequency components of solutions and disturbs capturing high-frequency or singular behaviors (Rahaman et al., 2019). Although some solutions to the spectral bias have been proposed (Tancik et al., 2020; Sitzmann et al., 2020), eliminating inductive biases from neural networks remains a challenge.

To address these issues, recent studies have explored combining classical grid-based representations with lightweight neural networks (Hui et al., 2018; Cao et al., 2023). In this approach, the parametric grids map input coordinates to intermediate features, which are then processed by neural networks to produce the final solutions. By relying on high-resolution parametric grids for representational capacity, this approach presents the potential to reduce the impact of neural networks' inductive biases. Moreover, using lightweight neural networks significantly reduces computational demands, leading to faster training speeds compared to traditional approaches only using neural networks.

While promising, existing methods that combine parametric grids with neural networks face a fundamental challenge. The positions of the parameters (the locations of vertices) are predetermined by the grid resolutions and remain fixed during training. Since the optimal allocation of representational capacity (determining where to place more vertices) is unknown, these methods typically use high-resolution grids that uniformly distribute many vertices across the entire input domain to achieve high accuracy. This approach results in using a large set of learnable parameters, which often leads to overfitting issues, i.e., low PDE residual losses but inaccurate solutions. To mitigate this problem, a large number of collocation points are sometimes used during training at the expense of the increased computational costs.

In this work, we introduce a novel representation for approximating solutions to PDEs. Drawing inspiration from adaptive mesh-based numerical methods (Berger & Oliger, 1984; Seol et al., 2016) and recent parametric grid representations (Li & Lee, 2021; Jang et al., 2023), we propose Physics-Informed Gaussian (*PIG*) that learns feature embeddings of input coordinates, using a mixture of Gaussian functions. For a given input coordinate, *PIG* extracts a feature vector as the weighted sum of the feature embeddings held by Gaussians with their learnable parameters (positions and shapes). They are adjusted during the training process, and underlying PDEs govern this dynamic adjustment. To update the parameters of all Gaussians, we leverage the well-established PINNs training framework, which employs numerous collocation points to compute PDE residuals and uses gradient-based optimization algorithms.

The proposed approach offers several advantages over existing parametric grid methods. *PIG* dynamically adjusts the computational mesh structure and the basis functions (Gaussians) to learn the feature embeddings. By following the gradient descent directions, the Gaussians move towards regions with high residual losses or singularities, and this adaptive strategy allows for more efficient

and precise solutions than the static uniform grid structures. In addition, Gaussian functions are infinitely differentiable everywhere, allowing for the convenient computation of high-order derivatives for PDE residuals, and they can be seamlessly integrated into deep-learning computation pipelines. The final architecture of the proposed approach, presented in Figure 2-(c), which combines the learnable Gaussian feature embedding and the lightweight neural network is a new learning-based PDE solver that can provide more efficient and accurate numerical solutions.

We have tested the proposed method on an extensive set of challenging PDEs (Raissi et al., 2019; Wang et al., 2021; Kang et al., 2023; Wang et al., 2023; Cho et al., 2024; Wang et al., 2024b). The experimental results show that the proposed *PIG* achieved competitive accuracy compared to the existing methods that use large MLPs or high-resolution parametric grids. When the number of Gaussians in *PIG* is comparable to the number of vertices in previous parametric grids, our method significantly outperformed existing approaches, demonstrating its superior efficiency. Furthermore, the proposed *PIG* shows significantly faster convergence speed than PINNs using large neural networks, demonstrating its effectiveness as a promising learning-based PDE solver. Our contributions are summarized as follows.

- We introduce Physics-Informed Gaussians, an efficient and accurate PDE solver that utilizes learnable Gaussian feature embeddings and a lightweight neural network.
- We propose a dynamically adaptive parametric mesh representation that effectively addresses the challenges encountered in previous static parametric grid approaches.
- We demonstrate that *PIG* achieves competitive accuracy and faster convergence with fewer parameters compared to state-of-the-art methods, establishing its effectiveness and paving the way for new research avenues.

## 2 RELATED WORK

### 2.1 PHYSICS-INFORMED NEURAL NETWORKS

PINNs are a class of machine learning algorithms designed to integrate physical laws into the learning process, popularized by Raissi et al. (2019). This is achieved by incorporating the PDE residuals directly into the loss function, allowing the model to be trained using standard gradient-based optimization methods. PINNs have gained significant attention for their ability to handle complex problems (Yang et al., 2021; Pensoneault & Zhu, 2024) including high-dimensional PDEs (Wang et al., 2022b; Hu et al., 2024b;a) that are challenging for traditional numerical methods. They are particularly effective in scenarios where data is sparse or expensive to obtain, as they can incorporate prior knowledge about the physical system. Applications of PINNs span various domains, including fluid dynamics, solid mechanics, and electromagnetics, demonstrating their versatility and effectiveness in solving real-world problems (Cai et al., 2021; Khan & Lowther, 2022; Bastek & Kochmann, 2023). Key advantages of PINNs include their mesh-free nature, the ability to easily incorporate boundary and initial conditions, and their flexibility in handling various types of PDEs. However, they also face challenges, such as the need for extensive computational resources and the difficulty in training deep networks to achieve accurate solutions. For example, Wang et al. (2024b) typically uses around 9 hidden layers with 256 hidden units (sometimes up to 18 layers) to achieve high accuracy. This requires massive computations to run the neural network, which involves multiple forward and backward passes to compute the gradients for PDE residual loss.

### 2.2 PHYSICS-INFORMED PARAMETRIC GRID REPRESENTATIONS

Physics-informed parametric grid representations combine traditional grid-based methods with neural networks to solve PDEs (Kang et al., 2023; Huang & Alkhalifah, 2024; Wang et al., 2024a; Shishehbor et al., 2024a). These representations have also been extensively explored in image, video, and 3D scene representations (Liu et al., 2020; Yu et al., 2021; Fridovich-Keil et al., 2022; Müller et al., 2022; Chen et al., 2022; Sun et al., 2022; Fridovich-Keil et al., 2023) by training the models as supervised regression problems. By discretizing the domain into a grid and associating each grid point with trainable parameters, these methods leverage the structured nature of grids to capture spatial variations effectively. This hybrid approach maintains high accuracy and reduces computational costs compared to purely neural network-based methods. Key benefits include the

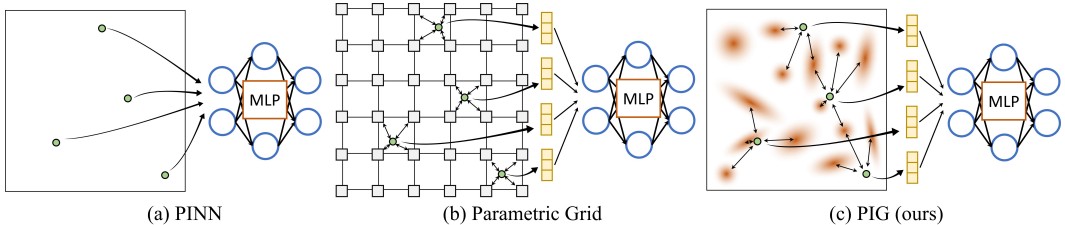

Figure 2: (a) PINN directly takes input coordinates (four collocation points) as inputs and produces outputs. (b) Parametric grids first map input coordinates to output feature vectors. Each vertex in the grids holds learnable parameters, and output features are extracted through interpolation schemes. (c) The proposed PIG consists of numerous Gaussians moving around within the input domain, and their shapes change dynamically during training. Each Gaussian has learnable parameters, and a feature vector for an input coordinate is the weighted sum of the learnable parameters based on the distance to the Gaussians.

ability to handle high-resolution representations and integrate boundary conditions efficiently, which are especially important for solving PDEs. However, the fixed grid structure can lead to suboptimal allocation of representational capacity during training.

## 2.3 ADAPTIVE MESH-BASED METHODS

Adaptive mesh-based methods dynamically adjust the computational mesh to minimize the error between approximated and true solutions. This process involves *a posteriori* error analysis, which estimates errors after solving, allowing for targeted mesh refinement. Such adaptivity is crucial in the numerical analysis as it ensures efficient allocation of computational resources, focusing on regions with high errors and thus improving overall accuracy and efficiency (Ainsworth & Oden, 1993; 1997).

There are also some studies on non-uniform adaptive sampling methods in the context of PINNs. Lu et al. (2021) proposed a residual-based adaptive refinement method in their work with Deep-XDE, aiming to enhance the training efficiency of PINNs (Wu et al., 2023). More recently, Yang et al. (2023b) introduced Dynamic Mesh-based Importance Sampling (DMIS), a novel approach that constructs a dynamic triangular mesh to efficiently estimate sample weights, significantly improving both convergence speed and accuracy. Similarly, Yang et al. (2023a) developed an end-to-end adaptive sampling framework called MMPDE-Net, which adapts sampling points by solving the moving mesh PDE. When combined with PINNs to form MS-PINN, MMPDE-Net demonstrated notable performance improvements. While these adaptive methods offer significant benefits, they also introduce additional complexity into the PINN framework.

## 2.4 POINT-BASED REPRESENTATIONS

Irregular point-based representations have long been considered promising approaches for data representation, reconstruction, and processing (Qi et al., 2017; Xu et al., 2022; Zhang et al., 2022). A recent study in 3D scene representation utilized Gaussians as a graphical primitive and showed remarkable performance in image rendering quality and training speed (Kerbl et al., 2023). The combination of Gaussian representation and neural networks has recently been explored in regressing images or 3D signed distance functions, showing its great expressibility (Chen et al., 2023). While those studies share some architectural similarities with our method, they all primarily focus on supervised regression problems to reconstruct the visual signals. We developed the architecture suitable for effective PDE solvers and first showed that the Gaussian features and neural networks can be trained in an unsupervised manner guided by the physical laws.

## 3 METHODOLOGY

### 3.1 PRELIMINARY: PHYSICS-INFORMED NEURAL NETWORKS

Consider an abstract underlying equation,

$$\mathcal{D}[u](x) = f(x), \quad x \in \Omega \subset \mathbb{R}^d, \tag{1}$$

$$\mathcal{B}[u](x) = g(x), \quad x \in \partial\Omega, \tag{2}$$

where $\mathcal{D}$ is a differential operator, and $\mathcal{B}$ is a boundary operator which could contain the initial condition. The physics-informed neural network methods try to find an approximate solution by minimizing

$$L(\theta) = \int_\Omega |\mathcal{D}[u_\theta](x) - f(x)|^2 dx + \lambda \int_{\partial\Omega} |\mathcal{B}[u_\theta](x) - g(x)|^2 d\sigma(x) \tag{3}$$

where $u_\theta$ is a neural network with the set of network parameters $\theta$, $\lambda$ is a positive real number, and $\sigma$ is a surface measure. In practice, integrals are usually estimated via Monte Carlo integration. PINNs typically utilize automatic differentiation to compute the PDE residuals and $\nabla_\theta L(\theta)$. For more details, please refer to the original paper (Raissi et al., 2019).

### 3.2 PHYSICS-INFORMED GAUSSIANS

In this section, we present the proposed Physics-Informed Gaussian representation (*PIG*) for numerical solutions to PDEs. It comprises two stages: Gaussian feature embedding (3.2.1) and feature refinement with a lightweight MLP (3.2.2).

### 3.2.1 LEARNABLE GAUSSIAN FEATURE EMBEDDING

Let $\phi = \{(\mu_i, \Sigma_i, f_i) : i = 1, \ldots, N\}$ be the set of Gaussian model parameters, where $\mu_i \in \mathbb{R}^d$ is a position of a Gaussian and $\Sigma_i \in \mathbb{S}^d_{++}$ is a covariance matrix. Each Gaussian has a learnable feature embedding $f_i \in \mathbb{R}^k$ for a feature dimension $k$. For simplicity, we consider $k = 1$. Given an input coordinate $x \in \mathbb{R}^d$, the learnable embedding $\mathtt{FE}_\phi : \mathbb{R}^d \to \mathbb{R}$ extracts Gaussian features as follows.

$$\mathtt{FE}_\phi(x) = \sum_{i=1}^N f_i G_i(x), \quad G_i(x) = e^{-\frac{1}{2}(x-\mu_i)^\top \Sigma_i^{-1}(x-\mu_i)}, \tag{4}$$

where $N$ is the number of Gaussians and $G_i$ represents the $i$-th Gaussian function. $\mathtt{FE}_\phi$ maps an input coordinate to a feature embedding by a weighted sum of the individual features $f_i$ of each Gaussian. Extensions to $k > 1$ for enhanced expressiveness are provided in Appendix A.1.

Gaussian features distant from the input coordinates do not contribute to the final feature embedding, while only neighboring Gaussian features remain significant. Similar to the previous parametric grid methods, which obtain feature embeddings by interpolating only neighboring vertices, this locality encourages the model to capture high-frequency details by effectively alleviating spectral bias.

All Gaussian parameters $\phi$ are learnable and iteratively updated throughout the training process. This dynamic adjustment, akin to adaptive mesh-based numerical methods, optimizes the structure of the underlying Gaussian functions to accurately approximate the solution functions. For example, Gaussians will migrate to the regions with high-frequency or singular behaviors that require more computational parameters, following the gradients $\frac{\partial L}{\partial \mu_i}$ (see Figure 1). Compared to the existing parametric grid approaches, which achieve this goal by uniformly increasing grid resolution, the proposed method can build a more parameter-efficient and optimal mesh structure.

### 3.2.2 LEARNABLE FEATURE REFINEMENT

Once the features are extracted, a neural network processes the feature to produce the solution outputs.

$$u_{\phi,\theta}(x) = \mathtt{NN}_\theta(\mathtt{FE}_\phi(x)), \tag{5}$$

where $\mathtt{NN}_\theta$ is a lightweight MLP with the parameter $\theta$. We employed a single hidden layer MLP with a limited number of hidden units, adding negligible computational costs. Feature extraction

plays a primary role in producing the final solution, while the MLP functions as a feature refinement mechanism. Even though Gaussian features are already universal approximators (see 3.3), using a small MLP at the end improved the solution accuracy by a large margin compared to the method without the MLP, i.e., $u_\phi(x) = \text{FE}_\phi(x)$.

### 3.2.3 PIG as a Neural Network

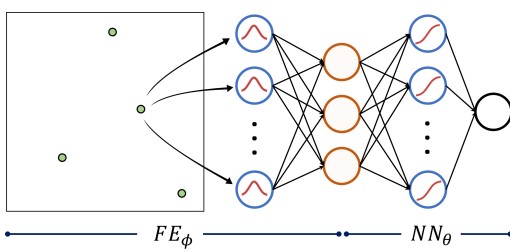

Figure 3: *PIG* as a neural network.

The proposed Gaussian feature embedding admits a form of radial basis function (RBF) network. Figure 3 depicts the overall *PIG* architecture as a neural network. The first layer contains $N$ (the number of Gaussians) RBF units, and an input coordinate passes through all RBF units, $G_i(x)$, resulting in a $N$-dimensional vector. A single fully connected layer processes this vector to produce a $k$-dimensional feature vector. The weight matrix $W \in \mathbb{R}^{k \times N}$ in this layer corresponds to the feature vectors held by each Gaussian, i.e., $W_{:,i} \in \mathbb{R}^k$ equals $f_i \in \mathbb{R}^k$.

The extracted feature vector is further processed by a single hidden layer MLP (we used the *tanh* activation function) to produce the final output, as depicted in Figure 3. Overall, *PIG* can be interpreted as an MLP with one input layer with $N$ RBF units and two hidden layers (no activation for the first hidden layer, and *tanh* for the second hidden layer).

A related study by Bai et al. (2023) has explored solving various PDEs using RBF networks (Park & Sandberg, 1991; Buhmann, 2000) within the framework of physics-informed machine learning. However, their approach differs from ours in that the positions of the basis functions are fixed. In contrast, our method allows the positions of the Gaussians to adjust dynamically, moving in directions that minimize the loss function. In addition, we extract the feature vectors from Gaussians and further process them using shallow neural networks while they directly predict the solution output from the Gaussians.

### 3.3 Universal Approximation Theorem for PIGs

Here, we present the Universal Approximation Theorem (UAT) for PIGs. A PIG consists of two functions: $\text{FE}_\phi$ and $\text{NN}_\theta$ (see equation 5). We will prove the UAT only for $\text{FE}_\phi$, as the UAT for PIGs follows directly from the standard UAT for MLPs. Given our earlier discussion on the relationship between PIGs and radial basis function networks, we begin with the following UAT specific to radial basis function networks.

**Theorem 1 ( Park & Sandberg (1991))** *Let $K : \mathbb{R}^d \to \mathbb{R}$ be an integrable bounded function such that $K$ is continuous and*

$$\int_{\mathbb{R}^d} K(x)\, dx \neq 0. \tag{6}$$

*Then the family $S_K$, defined as linear combinations of translations of $K$,*

$$S_K = \left\{ \sum_{i=1}^{n} f_i K(x - \mu_i) \middle| f_i \in \mathbb{R}, \mu_i \in \mathbb{R}^d, n \in \mathbb{N} \right\}, \tag{7}$$

*is dense in $C(\mathbb{R}^d)$.*

However, Theorem 1 does not apply to PIGs, as the feature embedding $\text{FE}_\phi$ in PIGs takes a slightly different form:

$$\text{FE}_\phi(x) = \sum_{i=1}^{n} f_i K\left(x - \mu_i; \Sigma_i\right), \tag{8}$$

where the key difference lies in the presence of $\Sigma_i$. Notably, the set

$$S'_K = \left\{ \sum_{i=1}^{n} f_i K(x - \mu_i; \Sigma_i) \middle| f_i \in \mathbb{R}, \mu_i \in \mathbb{R}^d, \Sigma_i \in \mathbb{S}^d_{++}, n \in \mathbb{N} \right\}, \tag{9}$$

| Methods | Allen-Cahn | Helmholtz | Nonlinear Diffusion | Flow Mixing | Klein-Gordon |
|---|---|---|---|---|---|
| PINN | - | 4.02e-1 | 9.50e-3 | - | 3.43e-2 |
| LRA | - | 3.69e-3 | - | - | - |
| PIXEL | 8.86e-3 | 8.63e-4 | - | - | - |
| SPINN | - | - | 6.10e-3 | 2.90e-3 | 3.90e-3 |
| JAX-PI | 5.37e-5 | - | - | - | - |
| PirateNet | **2.24e-5** | - | - | - | - |
| PIG (Ours) | 1.04e-4 | 4.13e-5 | 2.69e-3 | 4.51e-4 | 2.76e-3 |
| ± 1std | ± 4.12e-5, | ± 2.59e-05, | ± 6.55e-4, | ± 1.74e-4, | ± 4.27e-4, |
| best | 5.93e-5 | **2.12e-5** | **1.44e-3** | **2.67e-4** | **2.36e-3** |

Table 1: Comparison of relative $L^2$ errors across different methods. Three experiments were conducted using seeds 100, 200, and 300, with the mean and standard deviation presented in the table. The methods compared include PINN (Raissi et al., 2019), Learning Rate Annealing (LRA) (Wang et al., 2021), PIXEL (Kang et al., 2023), SPINN (Cho et al., 2024), JAX-PI (Wang et al., 2023), and Pirate-Net (Wang et al., 2024b). For fair comparisons, we included the reported values from the respective references and omitted results that were not provided in the original papers.

with $\mathbb{S}_{++}$ denoting the set of positive definite matrices, contains $S_K$. Therefore, $S'_K$ is dense in $C(\mathbb{R}^d)$. We summarize this in the following corollary:

**Corollary 1** *The scalar-valued, $d$-dimensional PIGs $\{NN_\theta \circ FE_\phi | (\theta, \phi) \in \mathbb{R}^{p_1 + p_2}\}$ are dense in $C(\mathbb{R}^d)$.*

## 4 EXPERIMENTS

### 4.1 EXPERIMENTAL SETUP

To validate the effectiveness of PIGs, we conducted extensive numerical experiments on various challenging PDEs, including Allen-Cahn, Helmholtz, Nonlinear Diffusion, Flow Mixing, and Klein-Gordon equations (for more experiments, please refer to the Appendix). We used the Adam optimizer (Kingma & Ba, 2014) for all equations except for the Helmholtz equation, in which the L-BFGS optimizer (Liu & Nocedal, 1989) was applied for a fair comparison to the baseline method PIXEL. For computational efficiency, we considered a diagonal covariance matrix $\Sigma = \text{diag}(\sigma_1^2, \ldots, \sigma_d^2)$ and we will discuss non-diagonal cases in Section 4.3.3.

### 4.2 EXPERIMENTAL RESULTS

#### 4.2.1 (1+1)D ALLEN-CAHN EQUATION

We compared our method against one of the state-of-the-art PINN methods on the Allen-Cahn equation, JAX-PI (Wang et al., 2023). For the detailed description, please refer to Appendix A.2.1. As shown in Figure 4, our method converges significantly faster and achieves competitive final accuracy (see Table 1). JAX-PI used a modified MLP architecture and 4 hidden layers with 256 hidden neurons. Thus, the number of parameters in JAX-PI is more than 250K, while ours used only around 20K parameters ($(N, d, k) = (4000, 2, 1)$). Also, note that the relative $L^2$ error curve in Figure 4 is displayed per iteration, and computational costs per iteration of ours are significantly lower than JAX-PI ($7.25 \times 10^{-3}$s/it vs. $1.67 \times 10^{-2}$s/it), which requires multiple forward and backward passes of the wide and deep neural network.

#### 4.2.2 2D HELMHOLTZ EQUATION

Figure 5 illustrates the numerical performance of our proposed PIG method for the 2D Helmholtz equation, comparing it to PIXEL (Kang et al., 2023), one of the state-of-the-art methods within the PINN family that uses parametric grid representations. A more detailed description of the experimental setup is available in Appendix A.2.2. The experiments were conducted using three different random seeds, with PIG achieving the best relative $L^2$ error of $2.22 \times 10^{-5}$ when employing the L-BFGS optimizer, and a relative $L^2$ error of $2.12 \times 10^{-5}$ with the Adam optimizer (For fair com-

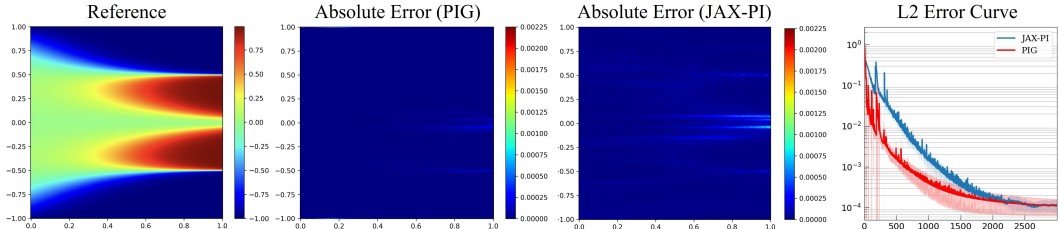

Figure 4: Allen-Cahn Equation. Reference solution and absolute error maps of PIG and one of the state-of-the-art methods (JAX-PI) to Allen-Cahn Equation (x-axis: $t$, y-axis: $x$). The rightmost depicts a relative $L^2$ error curve during the training process (x-axis: iterations, y-axis: $L^2$ error). The experiment was conducted with three different seeds, and the best relative $L^2$ error of PIG is $5.93 \times 10^{-5}$.

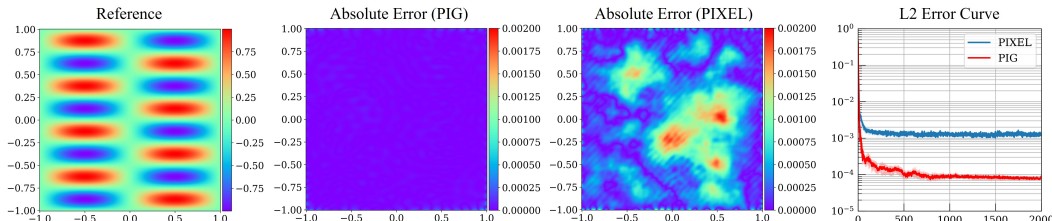

Figure 5: 2D Helmholtz Equation. Reference solution and absolute error maps of PIG and one of the state-of-the-art methods (PIXEL) to 2D Helmholtz Equation. The rightmost depicts a relative $L^2$ error curve during the training process and the best relative $L^2$ error of PIG is $2.22 \times 10^{-5}$.

parison, we reported the result using L-BFGS since PIXEL used L-BFGS). Notably, the results show that PIG's error is four times lower than that of PIXEL, highlighting the efficiency and accuracy of our method. We did not compare against other state-of-the-art methods, such as JAX-PI or Pirate-Net, as they did not conduct experiments in this setting. While we could have used their codes, the sensitivity of PINN variants to hyperparameters complicates fair comparisons.

### 4.2.3 (2+1)D KLEIN-GORDON EQUATION

Figure 6 presents the predicted solution profile for the Klein-Gordon equation, comparing our results with SPINN. The best relative $L^2$ error achieved is $2.36 \times 10^{-3}$. For further details, please refer to Appendix A.2.3.

### 4.2.4 (2+1)D NONLINEAR DIFFUSION EQUATION

We evaluated the performance of PIGs on the (2+1) dimensional nonlinear diffusion equation, with visualizations presented in Figure 18. The relative $L^2$ error achieved is $1.44 \times 10^{-3}$. For details on the experimental setup, please refer to Appendix A.2.5.

### 4.2.5 (2+1)D FLOW MIXING PROBLEM

Figure 7 displays the numerical solutions and absolute errors for the (2+1) flow mixing problem. Our solutions closely match the reference, with PIG achieving the relative $L^2$ error of $2.67 \times 10^{-4}$, compared to $2.90 \times 10^{-3}$ for SPINN, underlining the enhanced accuracy of PIG. Figure 19 presents solution profiles up to $t = 4$. Additional details can be found in Appendix A.2.4.

### 4.3 HYPERPARAMETER ANALYSIS AND ABLATION STUDY

In this section, we present the experimental results to show the effects of each component of the proposed PIG (using MLP, learnable Gaussian positions, and dense covariance matrices), In addition, we study the effect of the number of Gaussians, the size of MLP and input dimensions.

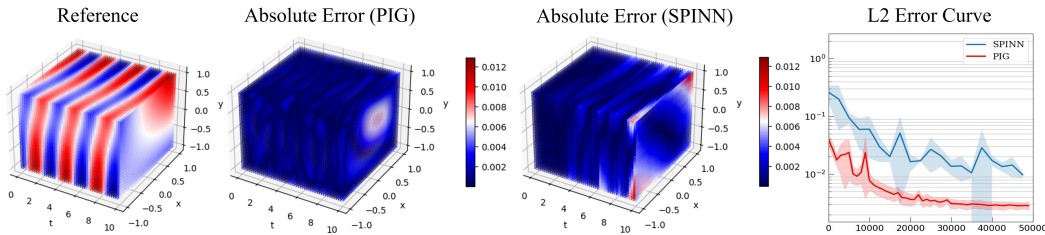

Figure 6: Klein-Gordon Equation. Reference solution and absolute error maps of PIG and one of the state-of-the-art methods (SPINN) to Klein-Gordon Equation. Both models used $16^3$ collocation points. The rightmost panel depicts a relative $L^2$ error curve during the training process and the best relative $L^2$ error of PIG is $2.36 \times 10^{-3}$.

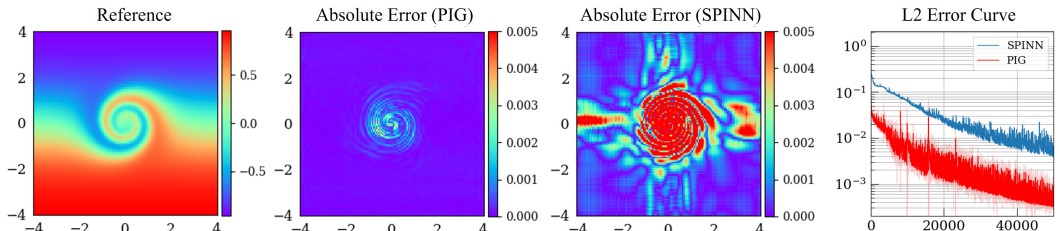

Figure 7: Flow mixing problem. The best relative $L^2$ error of PIG is $2.67 \times 10^{-4}$, while its maximum absolute error is $4.47 \times 10^{-3}$. In comparison, one of the state-of-the-art methods, SPINN achieved $2.90 \times 10^{-3}$ $L^2$ error and showed a maximum absolute error of $3.23 \times 10^{-2}$.

### 4.3.1 THE NUMBER OF GAUSSIANS

In numerical analysis, there is a general trend that the quality of the solution improves as the mesh is refined. Given our approach of using Gaussians as mesh points, we expect that the accuracy of PIGs will improve with an increased number of Gaussians. Table 2 illustrates the accuracy improvements of PIGs to the number of Gaussians. Overall, we observe a positive correlation between the number of Gaussians and improved accuracy. It is important to note that achieving this trend can be challenging for other PINN-type methods.

| # Gaussians | Flow Mixing | Nonlinear Diffusion | Allen-Cahn |
|---|---|---|---|
| 200 | 6.07e-03 | 2.33e-03 | 1.83e-02 |
| 400 | 3.13e-03 | 2.22e-03 | 2.93e-03 |
| 600 | 1.50e-03 | 2.23e-03 | 2.75e-03 |
| 800 | 1.44e-03 | 1.95e-03 | 1.22e-03 |
| 1000 | 1.31e-03 | 7.33e-03 | 4.81e-04 |
| 1200 | 1.03e-03 | 3.96e-03 | 3.98e-04 |

Table 2: The number of Gaussians and approximation accuracy (Flow Mixing, Nonlinear Diffusion, and Allen-Cahn). The results indicate that increasing the number of Gaussians typically leads to a decrease in relative $L^2$ error.

### 4.3.2 MLP IMPACT AND ADAPTIVE GAUSSIAN POSITIONS

While $\text{FE}_\phi$ serves as a universal approximator, we found that adding a small MLP $\text{NN}_\theta$ significantly enhances performance. Additionally, our ablation study explores the effectiveness of allowing adaptive Gaussian positions (learnable $\mu$ vs. fixed $\mu$). The results in Table 3 illustrate that varying Gaussian positions $\mu$ improve accuracy, particularly when combined with the MLP. We also evaluate the sensitivity of PIGs to the width and input dimensions of the MLP, as summarized in Table 8. Notably, no clear trend emerges, highlighting the robustness of PIGs to MLP variations.

| (MLP, $\mu$) | Allen-Cahn | Helmholtz | Nonlinear Diffusion | Flow Mixing | Klein-Gordon |
|---|---|---|---|---|---|
| (X, Fixed) | 4.72e-03 | 3.97e-04 | 6.32e-03 | 4.33e-03 | 6.44e-02 |
| (O, Fixed) | 1.82e-03 | 2.12e-04 | 2.10e-03 | 1.09e-03 | 2.69e-02 |
| (X, Learn) | 7.29e-05 | 1.86e-04 | 5.26e-03 | 7.93e-04 | 8.51e-03 |
| (O, Learn) | **7.27e-05** | **2.12e-05** | **1.44e-03** | **4.51e-04** | **2.76e-03** |

Table 3: Ablation study results on MLP and $\mu$ across various equations.

### 4.3.3 Covariance Matrices

Dense covariance matrices have the potential of improved accuracy over diagonal covariance matrices, at the expense of increased computational and memory costs. We compared these two types of covariance matrices across several equations: the 2D Helmholtz equation, the Klein-Gordon equation, the flow mixing problem, and the nonlinear diffusion equation. We found that training PIGs with dense covariance matrices from scratch did not result in accurate solutions. To address this issue, we first trained PIGs with diagonal covariance matrices and then initialized dense covariance matrices with pre-trained diagonal elements. As presented in Table 4, there were improvements in the Klein-Gordon and flow mixing equations. We believe that advanced training techniques and engineering would improve the performance of PIG with dense covariance matrices and leave it to future works.

| | Helmholtz | Klein-Gordon | Flow Mixing | Nonlinear Diffusion |
|---|---|---|---|---|
| Dense | 5.17e-05 | 1.81e-03 | 3.48e-04 | 3.86e-03 |
| Diagonal | 2.12e-05 | 2.76e-03 | 4.51e-04 | 1.44e-03 |

Table 4: Comparison of error levels between dense and diagonal covariance matrices in PIGs. For dense covariance matrix experiments, we first trained PIG using a diagonal covariance matrix and then fine-tuned full covariance matrix parameters initialized from the trained diagonal elements.

## 5 Conclusion and Limitations

In this work, we introduced *PIG*s as a novel method for approximating solutions to PDEs. By leveraging explicit Gaussian functions combined with deep learning optimization, *PIG*s address the limitations of traditional PINNs that rely on MLPs. Our approach dynamically adjusts the positions and shapes of the Gaussians during training, overcoming the fixed parameter constraints of previous methods and enabling more accurate and efficient numerical solutions to complex PDEs. Experimental results demonstrated the superior performance of PIGs across various PDE benchmarks, showcasing their potential as a robust tool for solving high-dimensional and nonlinear PDEs.

Despite the promising results, *PIG*s have certain limitations that warrant further investigation. Firstly, the dynamic adjustment of Gaussian parameters introduces additional computational overhead. While this improves accuracy, it may also lead to increased training times, particularly for very large-scale problems. However, by leveraging the locality of Gaussians, we can limit the evaluations to nearby Gaussians, which reduces the necessary computations and saves GPU memory. Secondly, the number of Gaussians is fixed at the beginning of training. Ideally, additional Gaussians should be allocated to regions requiring more computational resources to capture more complex solution functions. We believe it is a promising research direction and leave it to future work. Finally, a complete convergence analysis of the proposed method is not yet available. While empirical results show improved accuracy and efficiency, theoretical understandings of the convergence properties would provide deeper insights and guide further enhancements.

### Acknowledgments

This work was supported by the Basic Science Research Program through the National Research Foundation of Korea (NRF), funded by the Ministry of Education (NRF2021R1A2C1093579), the Korean government (MSIT) (RS-2023-00219980, RS-2024-00337548). This work was also supported by the Culture, Sports, and Tourism R&D Program through the Korea Creative Content

Agency grant funded by the Ministry of Culture, Sports and Tourism in 2024 (Project Name: Research on neural watermark technology for copyright protection of generative AI 3D content, RS-2024-00348469).

## REPRODUCIBILITY

We are committed to ensuring the reproducibility of our research. All experimental procedures, data sources, and algorithms used in this study are clearly documented in the paper. We already submitted the codes and command lines to reproduce the part of the results in Table 1 as supplementary materials. The code and datasets will be made publicly available upon publication, allowing others to validate our findings and build upon our work.

## ETHICS STATEMENT

This research adheres to the ethical standards required for scientific inquiry. We have considered the potential societal impacts of our work and have found no clear negative implications. All experiments were conducted in compliance with relevant laws and ethical guidelines, ensuring the integrity of our findings. We are committed to transparency and reproducibility in our research processes.

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

## A APPENDIX

### A.1 ENHANCED GAUSSIAN FEATURE EMBEDDING

To enhance the expressive capability, different Gaussians can be used for each feature dimension. The learnable Gaussian feature embedding $\text{FE}(x; \phi) : \mathbb{R}^d \to \mathbb{R}^k$ and the set of Gaussian model parameters $\phi = \{(\mu_i, \Sigma_i, f_i) : i = 1, \ldots, N\}$ are defined as previously described. Then, we have different Gaussians for each feature dimension, where $\mu_i \in \mathbb{R}^{k \times d}$ is the Gaussian position parameters, $\mu_{i,j} \in \mathbb{R}^d$ denotes the position parameter for $j$-th feature dimension, and $f_{i,j} \in \mathbb{R}$ represents $j$-th feature value. Similarly, $\Sigma_{i,j} \in \mathbb{S}_{++}^d$ is a covariance matrix for $j$-th feature dimension. Given an input coordinate $x \in \mathbb{R}^d$, $j$-th element of the learnable Gaussian feature embedding $\text{FE}_j(x; \phi)$ is defined as follows,

$$\text{FE}_j(x; \phi) = \sum_{i=1}^{N} f_{i,j} G_{i,j}(x), \quad G_{i,j}(x) = e^{-\frac{1}{2}(x - \mu_{i,j})^\top \Sigma_{i,j}^{-1}(x - \mu_{i,j})}, \tag{10}$$

where $G_{i,j}$ is the Gaussian function using Gaussians parameters for $j$-th feature dimension.

## A.2 DETAILED DESCRIPTION OF EXPERIMENTS

### A.2.1 (1+1)D ALLEN-CAHN EQUATION

The Allen-Cahn equation is a one-dimensional time-dependent reaction-diffusion equation that describes the evolutionary process of phase separation, which reads

$$u_t - 0.0001u_{xx} + 5u^3 - 5u = 0, \quad (x,t) \in [-1,1] \times [0,1], \tag{11}$$

with the periodic boundary condition

$$u(-1,t) = u(1,t), \quad u_x(-1,t) = u_x(1,t). \tag{12}$$

The initial condition for the experiment was $u(x,0) = x^2\cos(\pi x)$. We used the NTK-based loss balancing scheme (Wang et al., 2022a) to mitigate the ill-conditioned spectrum of the neural tangent kernel (Jacot et al., 2018). We used $N = 4000$ Gaussians for training and a diagonal covariance matrix for parameter efficiency, where the diagonal elements of the initial $\Sigma$ were set to a constant value of 0.025. The $\mu_i$ was uniformly initialized following $\text{Uniform}[0,2]^2$. We used shallow MLP with one hidden layer with 16 hidden units, and the dimension of the Gaussian feature was $k = 1$.

Reference solution was generated by Chebfun (Driscoll et al., 2014), which utilizes the Fourier collocation method with $N = 4096$ Fourier modes with ETDRK4 time stepping (Kassam & Trefethen, 2005) with a fixed time step $\Delta t = 1/200$.

### A.2.2 2D HELMHOLTZ EQUATION

The Helmholtz equation is the eigenvalue problem of the Laplace operator $\Delta = \nabla^2$. We consider the manufactured solution

$$u(x,y) = \sin(a_1\pi x)\sin(a_2\pi y), \quad (a_1,a_2) = (4,1), \tag{13}$$

to the two-dimensional Helmholtz equation with the homogeneous Dirichlet boundary condition given by

$$\Delta u + k^2 u = q, \quad (x,y) \in [-1,1]^2, \quad k = 1, \tag{14}$$

where

$$q(x,y) = k^2 \sin(a_1\pi x)\sin(a_2\pi y) - (a_1\pi)^2 \sin(a_1\pi x)\sin(a_2\pi y) - (a_2\pi)^2 \sin(a_1\pi x)\sin(a_2\pi y) \tag{15}$$

can be extracted from the solution $u$.

We used $N = 3000$ Gaussians in this experiment. The weights and scales of Gaussians were initialized following $\text{Uniform}[-1,1]$ and 0.1, respectively. The feature size of Gaussians was fixed at 4. The shallow MLP has 16 hidden nodes, and its network parameters were initialized by Glorot normal. The inputs for the Gaussians were rescaled into $[0,1]^2$, therefore the positions were initialized following $\text{Uniform}[0,1]^2$.

### A.2.3 (2+1)D KLEIN-GORDON EQUATION

The Klein-Gordon equation is a relativistic wave equation, which predicts the behavior of a particle at high energies. We consider the manufactured solution

$$u(x,y,t) = (x+y)\cos(2t) + xy\sin(2t) \tag{16}$$

to the (2+1) dimensional inhomogeneous Klein-Gordon equation

$$u_{tt} - \Delta u + u^2 = f, \quad (x,y,t) \in [-1,1]^2 \times [0,10], \tag{17}$$

where the forcing $f$, initial condition, and Dirichlet boundary condition are extracted from the manufactured solution $u$. In this experiment, we employed $N = 100$ Gaussians and a shallow MLP whose input dimension is 4 and hidden layer size is 16. The network parameters for the shallow MLP were initialized by Glorot Normal. Every weight of Gaussian was initialized following $\text{Normal}(0,0.01^2)$. The scale parameter $\sigma_i$'s were initialized with a constant value of 0.5. Instead of direct usage of the computational domain $[-1,1]^2 \times [0,10]$, we used linearly rescaled values $\in [0,2]^3$ for the inputs of Gaussians. Accordingly, position parameters of Gaussians were initialized following $\text{Uniform}[0,2]^3$.

### A.2.4 (2+1)D FLOW MIXING PROBLEM

A mixing procedure of two fluids in a two-dimensional spatial domain could be described in the following equation

$$u_t + au_x + bu_y = 0, \quad (x, y, t) \in [-4, 4]^2 \times [0, 4], \tag{18}$$

$$a(x, y) = -\frac{v_t}{v_{t,\max}} \frac{y}{r}, \quad b(x, y) = \frac{v_t}{v_{t,\max}} \frac{x}{r}, \tag{19}$$

$$v_t = \text{sech}^2(r)\tanh(r), \tag{20}$$

$$r = \sqrt{x^2 + y^2}, \quad v_{t,\max} = 0.385. \tag{21}$$

The analytic solution is $u(x, y, t) = -\tanh\left(\frac{y}{2}\cos(wt) - \frac{x}{2}\sin(wt)\right)$, where $w = v_t/(rv_{t,\max})$; see e.g., Tamamidis & Assanis (1993). The initial condition can be extracted from the analytic solution.

To predict the solution to the PDE, we used $N = 4000$ Gaussians. The weights and scales were initialized to $\text{Normal}(0, 0.01^2)$ and $0.1$, respectively. The size of Gaussian features was fixed at $4$. MLP had 16 hidden nodes, and its parameters were initialized by Glorot normal. Inputs for the Gaussians were rescaled to $[0, 2]^3$, hence the positions of Gaussians were initialized following $\text{Uniform}[0, 2]^3$.

### A.2.5 (2+1)D NONLINEAR DIFFUSION EQUATION

The diffusion equation is a parabolic PDE describing the diffusion process of a physical quantity, such as heat. We consider a nonlinear diffusion equation for our benchmark, which reads

$$u_t = 0.05 \left( \|\nabla u\|^2 + u\Delta u \right), \quad (x, y, t) \in [-1, 1]^2 \times [0, 1], \tag{22}$$

$$u_0(\mathbf{x}) = 0.25g\left(\mathbf{x}; 0.2, 0.3, \frac{1}{\sqrt{10}}\right) + 0.4g\left(\mathbf{x}; -0.1, -0.5, \frac{1}{\sqrt{15}}\right) + 0.3g\left(\mathbf{x}; -0.5, 0, \frac{1}{\sqrt{20}}\right),$$

where

$$\mathbf{x} = (x, y) \quad \text{and} \quad g(x, y; a, b, \sigma) = e^{-\frac{(x-a)^2 + (y-b)^2}{\sigma^2}}.$$

There are three peaks at the initial time and the peaks spread out as time goes on.

We employed $N = 4000$ Gaussians. The weights and scales of Gaussians were initialized to $\text{Normal}(0, 0.01^2)$ and $0.1$, respectively. The size of Gaussian features was $4$. The hyperbolic tangent MLP had only a single hidden layer with 16 nodes, and its parameters were initialized by Glorot normal. The inputs for the Gaussians were rescaled into $[0, 1]^3$. Correspondingly, the positions of Gaussians were initialized following $\text{Uniform}[0, 1]^3$.

### A.3 ADDITIONAL EXPERIMENTS

Here, we compare PIGs to PIRBNs (Bai et al., 2023). Two equations in the PIRBN paper are chosen as benchmarks.

Equation (15) in Bai et al. (2023):

$$\frac{\partial^2}{\partial x^2} u(x - 100) - 4\mu^2\pi^2 \sin(2\mu\pi(x - 100)) = 0, \tag{23}$$

and $u(100) = u(101) = 0$. The exact solution is $u(x) = -\sin(2\mu\pi(x - 100))$. We considered $\mu = 4$.

Equation (30) in Bai et al. (2023):

$$\begin{aligned}
\frac{\partial^2}{\partial x^2} u(x) = {}& -2\pi(22 - x)\cos(2\pi x) + 0.5\sin(2\pi x) - \pi^2(22 - x)^2 \sin(2\pi x) \\
& + 16\pi(x - 20)\cos(16\pi x) + 0.5\sin(16\pi x) - 64\pi^2(x - 20)^2 \sin(16\pi x),
\end{aligned} \tag{24}$$

and $u(20) = u(22) = 0$. The exact solution is $u(x) = \left(\frac{22-x}{2}\right)^2 \sin(2\pi x) + \left(\frac{x-20}{2}\right)^2 \sin(16\pi x)$.

Referring to the numbers in 5, PIGs achieved error levels by two orders of magnitude smaller than PIRBNs. This improvement could be attributed to the introduction of a tiny MLP and letting positions move during training.

|        | Equation 23            | Equation 24            |
|--------|------------------------|------------------------|
| PIRBNs | 6.87e-03 ± 3.70e-04    | 1.47e-02 ± 9.16e-03    |
| PIGs   | 1.79e-05 ± 3.80e-06    | 1.14e-04 ± 1.19e-05    |

Table 5: Results of the comparison study between PIGs and PIRBNs for Equations 23 and 24. PIGs achieve lower errors than PIRBNs, highlighting their superior performance in both equations.

### A.4 SEPARABLE PIGS

Separable PINNs have shown excellent performance across various PDEs (Cho et al., 2024; Oh et al., 2024). When mesh points are tensor products of 1D grids, the number of network forward passes of SPINNs scale linearly $O(Nd)$, in contrast to the exponential scaling $O(N^d)$ of traditional PINNs, which adopt a single MLP.

Here, we provide a proof-of-concept for combining SPINNs and PIGs. Separable PIGs (SPIGs) might have the following form:

$$u(x_1, \ldots, x_d) \approx \sum_{r=1}^{R} \prod_{i=1}^{d} \mathrm{PIG}_r(x_i; \theta_i) \tag{25}$$

where $\mathrm{PIG}_r$ is the $r$-th component of the output vector.

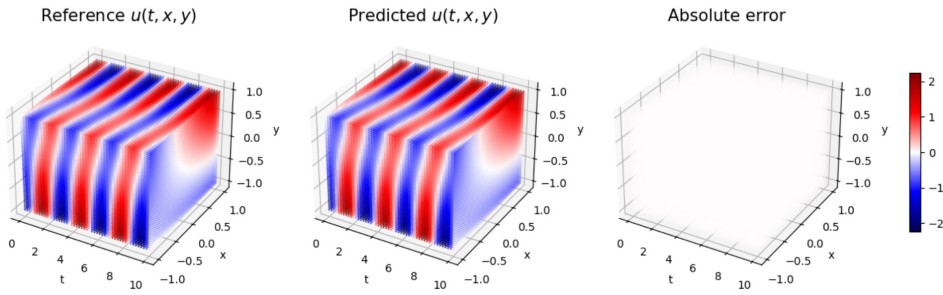

Figure 8: Klein-Gordon equationA.2.3. The relative $L^2$ error of SPIG is $3.68 \times 10^{-4}$.

**2D L-shaped Poisson equation** the two-dimensional Poisson equation defined on an L-shaped domain. Despite the non-tensor-product nature of the computational domain, SPINNs can deal with such complex domains by masking outputs. Please refer to (Cho et al., 2024) for the description of this benchmark problem. A SPIG achieved $1.89 \times 10^{-2}$ relative $L^2$ error for this problem, while SPINN solution was $3.22 \times 10^{-2}$.

**(2+1)D Klein-Gordon equation** SPIG achieved $3.68 \times 10^{-4}$ relative $L^2$ error. PIG's best relative $L^2$ error was $2.36 \times 10^{-3}$. Please refer to A.2.3 for a description of PDE. SPIG used modified MLP with 2 layer and 16 hidden features. The weights and scales were initialized to $\mathrm{Normal}(0, 0.01^2)$ and 0.1, respectively. position parameters of Gaussians were initialized following $\mathrm{Uniform}[0, 2]^3$. 2500 Gaussians are used.

**(3+1)D Klein-Gordon equation** SPIG achieved $2.88 \times 10^{-4}$ relative $L^2$ error. SPINN's relative $L^2$ error was $1.20 \times 10^{-3}$. Please refer to (Cho et al., 2024) for the description of this benchmark problem. SPIG used modified MLP with 2 layer and 16 hidden features. The weights and scales were initialized to $\mathrm{Normal}(0, 0.01^2)$ and 0.25, respectively. position parameters of Gaussians were initialized following $\mathrm{Uniform}[0, 2]^3$. 2500 Gaussians are used.

**3D Helmholtz equation** SPIG achieved $1.50 \times 10^{-3}$ relative $L^2$ error. SPINN's relative $L^2$ error was $3.00 \times 10^{-2}$. Please refer to (Cho et al., 2024) for the description of this benchmark problem. SPIG used modified MLP with 2 layers and 16 hidden features. The weights and scales were initialized to $\mathrm{Normal}(0, 0.01^2)$ and 0.05, respectively. position parameters of Gaussians were initialized following $\mathrm{Uniform}[0, 2]^3$. 2500 Gaussians are used.

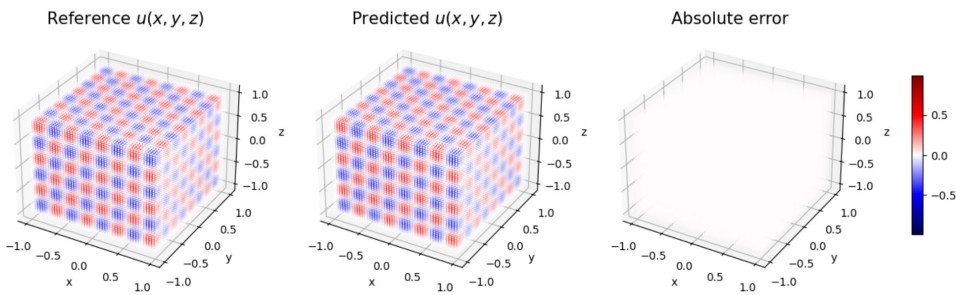

Figure 9: 3D Helmholtz equation A.4. The relative $L^2$ error of SPIG is $1.50 \times 10^{-3}$.

## A.5 INVERSE PROBLEM

With observation data, the PINN framework can estimate unknown equation parameters by letting them be learnable. Here we consider (1+1)D Allen-Cahn equation

$$u_t - 10^{-4}u_{xx} + \lambda u^3 - 5u = 0,$$

with an unknown coefficient $\lambda$. Other conditions are the same with Section A.2.1. We estimated $\lambda$ using reference solution as observation data. Figure 10 presents estimated $\lambda$ over iterations, clearly showing PIG's faster convergence.

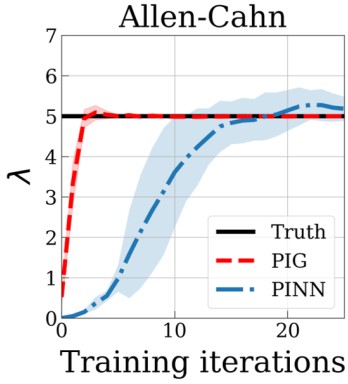

Figure 10: Allen-Cahn Inverse problem. The experiment was conducted on five different seeds (100, 200, 300, 400, 500). PIG showed better performance than PINN.

## A.6 HIGH-DIMENSIONAL EQUATIONS

Hu et al. introduced stochasticity in the dimension during the gradient descent (SDGD) to efficiently handle high-dimensional PDEs within the PINN framework Hu et al. (2024b). PIGs can utilize SDGD to tackle extremely high dimensional PDEs, e.g., 100D Allen-Cahn, and Poisson equation. Specifically, let $d = 100$ and $B^d = \{x \in \mathbb{R}^d : \|x\|_2 \le 1\}$ be the domain. We consider a function

$$u_{\text{exact}} = \left(1 - \|x\|_2^2\right) \left(\sum_{i=1}^{d-1} c_i \sin\left(x_i + \cos(x_{i+1}) + x_{i+1}\cos(x_i)\right)\right),$$

as our exact solution, where $c_i \sim \text{Normal}(0, 1^2)$. Our benchmark problems are the Poisson equation and the Allen-Cahn equation, which read

$$\Delta u = g \quad \text{(Poisson)} \quad \text{and} \quad \Delta u + u - u^3 = g \quad \text{(Allen-Cahn)}$$

where $g$ is induced from the exact solution.

Figure 11 presents relative $L^2$ error curves over iterations. Note that global polynomial-based methods cannot handle such high dimensional equations due to the curse of dimensionality.

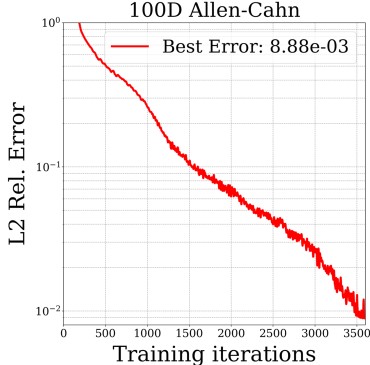 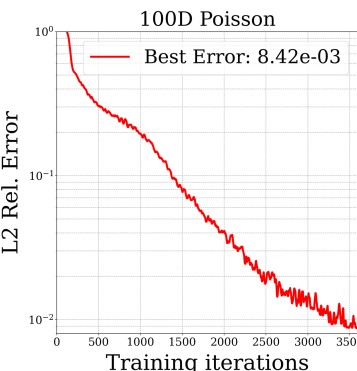

Figure 11: Relative $L^2$ error curves for two high dimensional PDEs. Left: 100D Allen-Cahn equation. Right: 100D Poisson equation. PIGs achieved $8.88 \times 10^{-3}$, and $8.42 \times 10^{-3}$, respectively.

### A.7 LID-DRIVEN CAVITY

To further illustrate the effectiveness of PIGs over traditional parametric mesh methods, we chose PGCAN Shishehbor et al. (2024b) as our baseline and considered the lid-driven cavity problem presented in the paper. The domain is $[0, 1]^2$. The homogeneous Dirichlet boundary condition is imposed except for the lid $\{(x, 1) : x \in [0, 1]\}$. The governing equation is a 2D stationary incompressible Naiver-Stokes equation,

$$\nabla \cdot u = 0$$
$$\rho(u \cdot \nabla)u = -\nabla p + \mu \nabla^2 u$$

where the boundary conditions are given as follows:

$$u(0, y) = u(1, y) = (0, 0),$$
$$u(x, 0) = (0, 0),$$
$$u(x, 1) = (A \sin(\pi x), 0), \quad A = 1,$$
$$p(0, 0) = 0.$$

We used 2000 Gaussians. Covariance matrices were diagonal and initialized at $0.1$ and positions were initialized following $\text{Uniform}[0, 2]$.

Figure 12 depicts numerical results. PIG shows excellent agreement with the reference solution. Figure 13 illustrates faster convergence of PIGs compared to the baseline method PGCAN.

### A.8 EXAMPLE FOR SPECTRAL BIAS

Figure 14 illustrates PIG's ability to approximate high-frequency functions. We considered 2D Helmholtz equation (see Section A.2.2) with a high wavenumber $(a_1, a_2) = (10, 10)$ for a benchmark problem.

### A.9 THE HISTOGRAM OF VARIANCES AND DISTANCES OF GAUSSIANS

Figure 15 shows the histograms of the Gaussian parameters for the two benchmark problems discussed in Section 4.2.5 and Section 4.2.3. Readers may observe that the Gaussians in the right panels are more global and, therefore, more sparsely distributed compared to those in the left panels.

### A.10 COMPARISON BETWEEN PIG AND SIREN

In this section, we compare the performance of PIG with SIREN Sitzmann et al. (2020). PIG is composed of a feature embedding $\text{FE}_\phi$ and a lightweight neural network $\text{NN}_\theta$. Here, we investigate the effectiveness of SIREN when used either as a feature embedding or as a lightweight neural network.

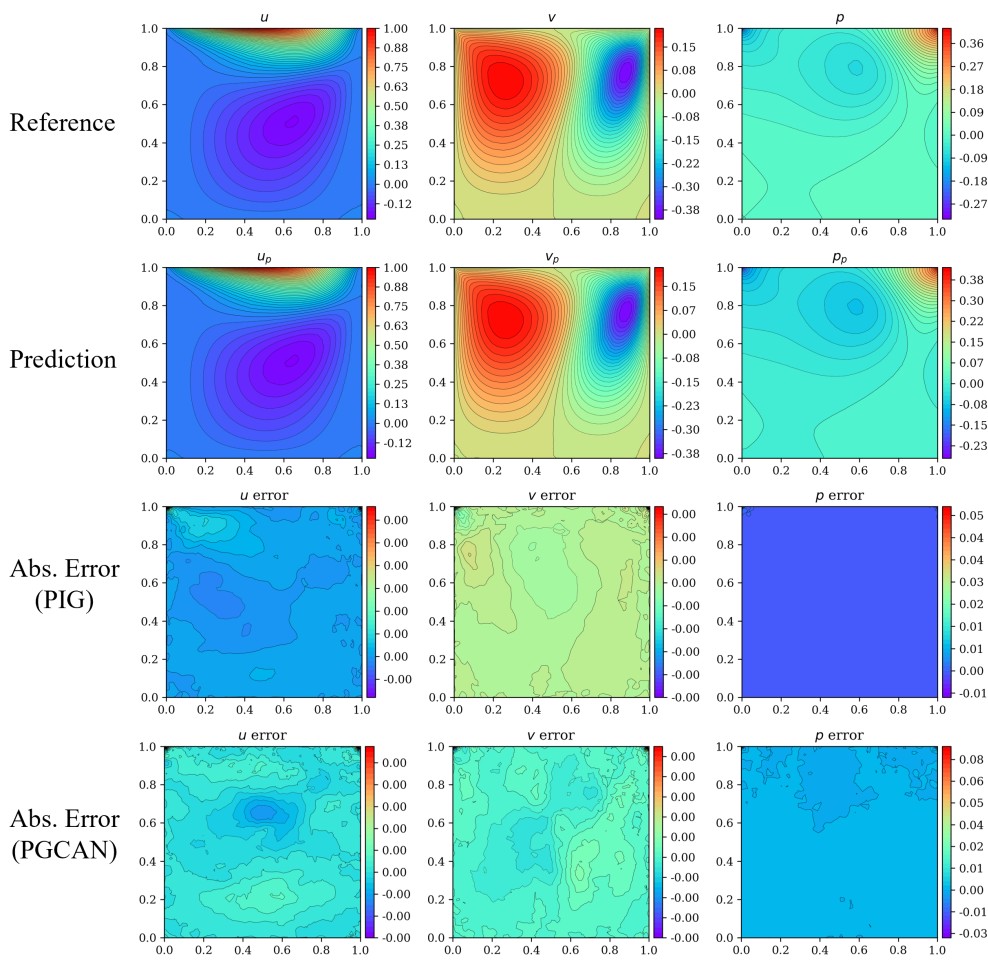

Figure 12: Lid-driven cavity flow problem. PIG achieved $4.04 \times 10^{-4}$ relative $L^2$ error whereas the baseline parametric grid method PGCAN resulted in $1.22 \times 10^{-3}$.

When used as $\mathrm{FE}_\phi$, SIREN is implemented as an MLP with 4 layers, each containing 256 units, and $\sin(3x)$ as the activation function. As $\mathrm{NN}_\theta$, SIREN is a shallow MLP with 16 units and $\sin(30x)$ activation function. It is worth noting that using $\sin(30x)$ as the activation function for the feature embedding $\mathrm{FE}_\phi$ did not yield effective results.

| $\mathrm{FE}_\phi + \mathrm{NN}_\theta$ | Helmholtz | Flow Mixing | Klein-Gordon |
|---|---|---|---|
| SIREN + Id | 1.68e-03 ± 2.02e-03 | 1.22e-02 ± 4.17e-03 | 1.18e-01 ± 4.88e-02 |
| SIREN + tanh | 1.31e-03 ± 8.26e-04 | 2.80e-02 ± 2.50e-02 | 1.04e-01 ± 8.61e-02 |
| PIG + SIREN | **1.37e-05** ± 1.64e-06 | 1.28e-03 ± 1.09e-04 | 2.37e-02 ± 4.62e-03 |
| PIG + tanh | 4.13e-05 ± 2.59e-05 | **4.51e-04** ± 1.74e-04 | **2.76e-03** ± 4.27e-04 |

Table 6: Comparison of PIG and SIREN performance. For all cases except the Helmholtz equation, the original PIG + tanh formulation outperformed other methods. The improved performance of PIG + SIREN on the Helmholtz equation may be attributed to the functional form of its exact solution.

The results, summarized in Table 6, indicate that SIREN as $\mathrm{FE}_\phi$ did not perform notably well. However, when SIREN was employed as $\mathrm{NN}_\theta$, it demonstrated excellent performance in solving the Helmholtz equation discussed in Section 4.2.2. This improvement is likely due to the structural similarity between the SIREN activation and the functional form of the exact solution (equation 13).

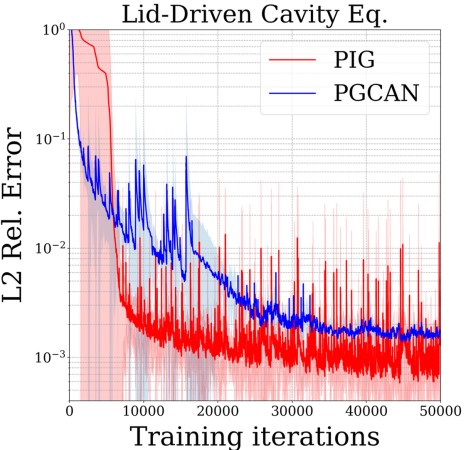

Figure 13: Relative $L^2$ error curve of the lid-driven cavity problem. PIG achieved $4.04 \times 10^{-4}$ and PGCAN which used the parametric grid method achieved $1.22 \times 10^{-3}$.

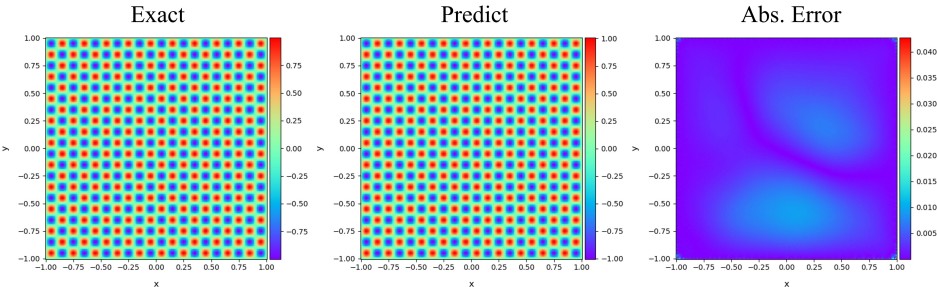

Figure 14: 2D Helmholtz equation with a high wavenumber $(a_1, a_2) = (10, 10)$. PIG achieved a relative $L^2$ error of $7.09 \times 10^{-3}$, while the parametric fixed grid method PIXEL reached a relative $L^2$ error of $7.47 \times 10^{-2}$. PINN failed to converge.

### A.11    COMPARISON WITH A CONCURRENT WORK

We conducted several experiments to compare PIG with the Physics-Informed Gaussian Splatting (PI-GS) method proposed by Rensen et al. (2024). Table 7 summarizes the relative $L^2$ errors and computation times per iteration (shown in parentheses). Across the three benchmark problems, PIG consistently outperforms PI-GS in both accuracy and efficiency.

|        | Burgers' equation (1)          | Burgers' equation (2)          |
|--------|--------------------------------|--------------------------------|
| PIG    | $7.68 \times 10^{-4}$ (0.28s/it) | $1.08 \times 10^{-3}$ (0.29s/it) |
| PI-GS  | $1.62 \times 10^{-1}$ (1.5s/it)  | $2.61 \times 10^{-1}$ (1.68s/it) |

Table 7: Performance comparison of PIG and PI-GS across 3 benchmark problems. Results include relative $L^2$ errors and computation times per iteration (s/it). Benchmarks are conducted on two variations of the (2+1)D Burgers equation.

Across all experiments, We considered $[-2.5, 2.5]^2$ as spatial domain with zero Dirichlet boundary condition.

**(2+1)D Burgers' Equation** For $\nu = 1/(10\pi)$, the (2+1)D Burgers' equation is given by:

$$\frac{\partial u}{\partial t} + u \frac{\partial u}{\partial x} = \nu \left( \frac{\partial^2 u}{\partial x^2} + \frac{\partial^2 u}{\partial y^2} \right),$$

where $u$ is a scalar-valued function.

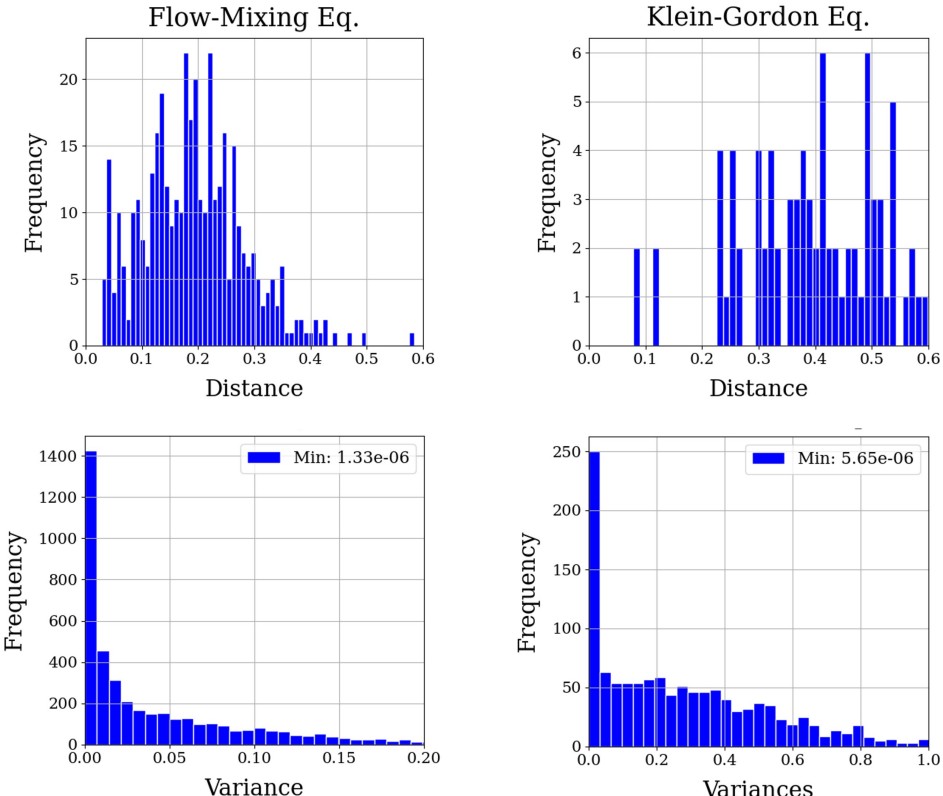

Figure 15: Histograms of the Gaussian parameters for the flow mixing problem and the Klein-Gordon equation. The upper panels display histograms of the minimum distances between the Gaussian centers, where distances $> 0$ indicate the absence of mode collapse. The lower panels show histograms of the Gaussian variances, highlighting the non-degeneracy of the Gaussians.

In the first example, the initial condition is set as the probability density function (PDF) of a standard two-dimensional Normal distribution. In the second example, the initial condition is the PDF of a mixture of two Gaussians. Figures 16 and 17 illustrate the reference solutions and the corresponding PIG solutions.

## A.12 ADDITIONAL FIGURES AND TABLES

| # Hidden units | MLP input dimension ($k$) | | | |
|---|---|---|---|---|
| | 1 | 2 | 3 | 4 |
| 4 | 7.77e-03 | 9.60e-03 | 7.68e-03 | 9.60e-03 |
| 8 | 8.55e-03 | 6.44e-03 | 1.06e-02 | 8.54e-03 |
| 16 | 8.24e-03 | 1.06e-02 | 1.21e-02 | 6.90e-03 |
| 32 | 7.14e-03 | 8.06e-03 | 1.22-02 | 6.87e-03 |
| 64 | 6.33e-03 | 7.50e-03 | 1.09e-02 | 9.48e-03 |
| 128 | 6.38e-03 | 6.88e-03 | 8.48e-03 | 7.47e-03 |
| 256 | 5.21e-03 | 6.60e-03 | 5.22e-03 | 5.40e-03 |

Table 8: The performance of different MLP configurations for the Helmholtz equation, displaying $L^2$ relative errors at iteration 1,000 across various configurations of hidden units and MLP input dimensions. Overall, the results highlight the robustness to the size of MLP, showing minimal variation in errors across different settings.

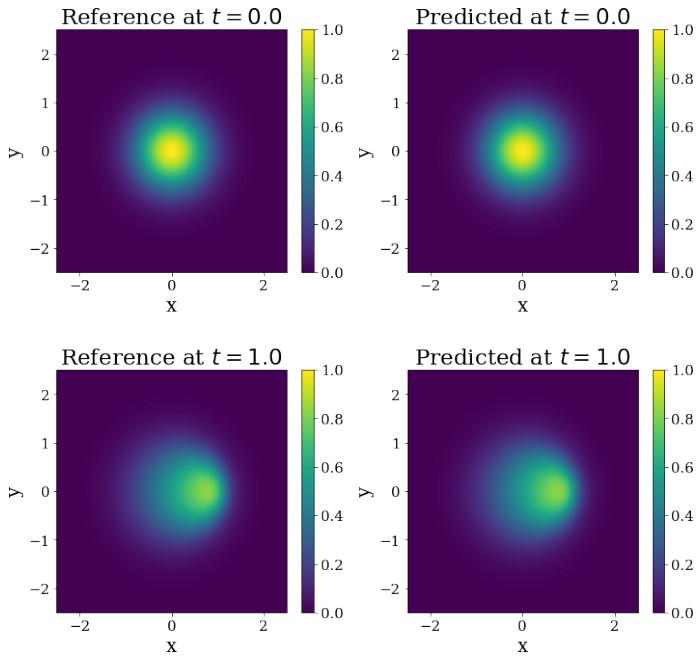

Figure 16: Prediction results of PIG for the first example of the (2+1)D Burgers' equation. PIG achieved a relative $L^2$ error of $7.68 \times 10^{-4}$, with a computation time of 0.28 seconds per iteration. In contrast, PI-GS attained a relative $L^2$ error of $1.62 \times 10^{-1}$, requiring 1.50 seconds per iteration.

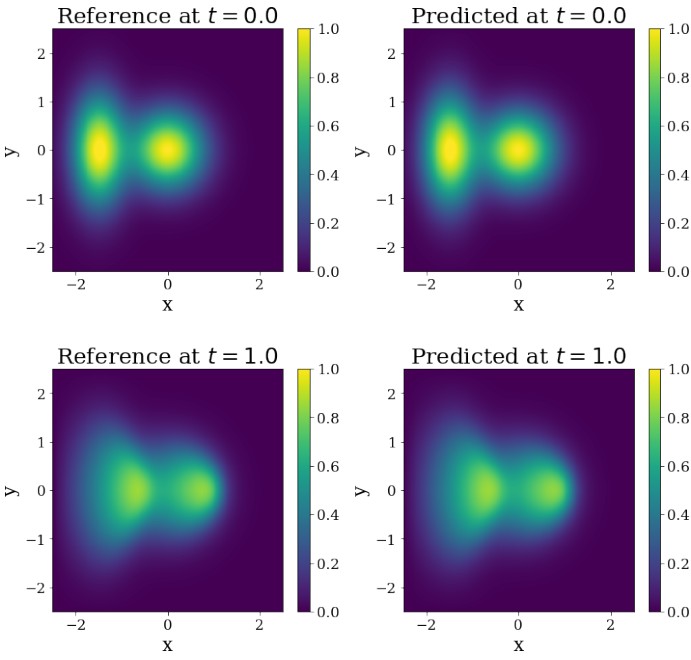

Figure 17: Prediction results of PIG for the second example of the (2+1)D Burgers' equation. PIG achieved a relative $L^2$ error of $1.08 \times 10^{-3}$, with a computation time of 0.29 seconds per iteration. In comparison, PI-GS attained a relative $L^2$ error of $2.61 \times 10^{-1}$, requiring 1.68 seconds per iteration.

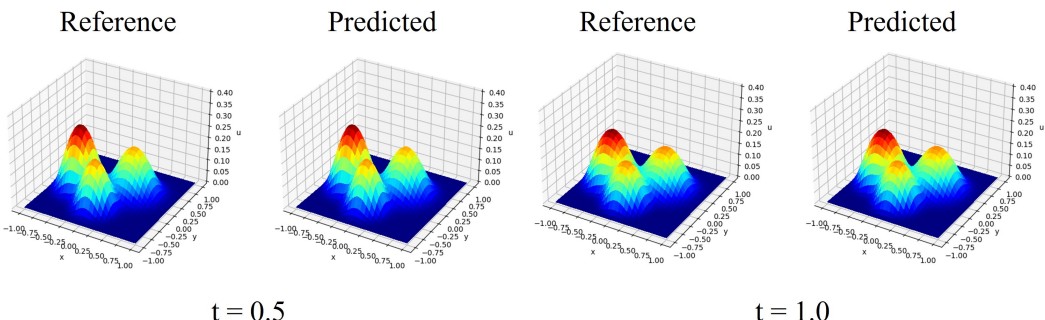

Figure 18: Non-linear diffusion equation 4.2.4. The experiment was conducted on three different seeds (100, 200, 300). The best relative $L^2$ error is $1.44 \times 10^{-3}$.

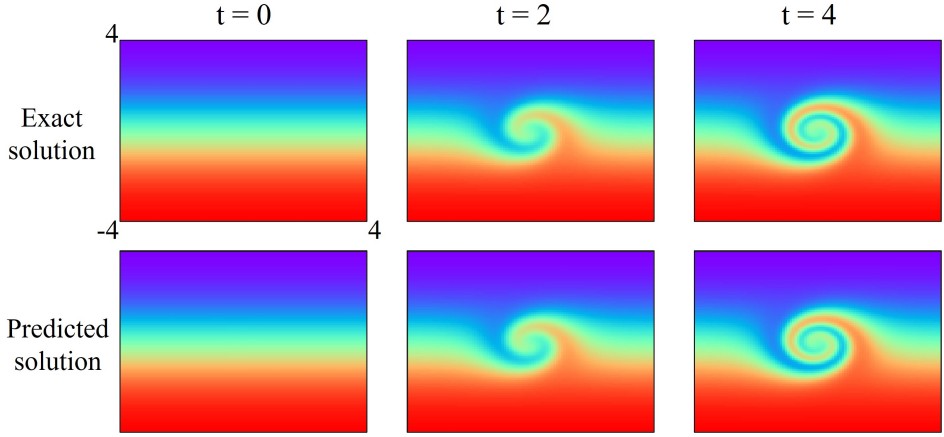

Figure 19: Flow mixing equation 4.2.5. The experiment was conducted on three different seeds (100, 200, 300). The best relative $L^2$ error is $2.67 \times 10^{-4}$.

