# OpenReview forum: "PIG: Physics-Informed Gaussians as Adaptive Parametric Mesh Representations"
_ICLR.cc/2025/Conference — ICLR 2025 Poster_

### Official Review · Reviewer_uWK5 · 2024-10-16

**Soundness:** 3
**Presentation:** 3
**Contribution:** 3
**Rating:** 6
**Confidence:** 2

**Summary:**

This paper proposes Physics-Informed Gaussians (PIGs), which incorporates Gaussian embeddings in the PINNs framework, to eliminate the inductive biases of neural networks.

**Strengths:**

The strength of this paper is to overcomes the limitations of traditional PINNs.  The proposed method maintains the benefits of PINNs, improves performance in PDE approximation, and can be applied to various PDEs.

**Weaknesses:**

As mentioned in Discussion and Limitations, theoretical understanding of the convergence properties is lack, although the universality of PIG is discussed in Section 3.3.

**Questions:**

I may miss something but, is there any reason why choose Gaussian functions as feature embedding? Would other embedding functions fail in the proposed framework?

---

> ### Author Response · Authors · 2024-11-22
>
> ### **W1. Theoretical understanding of the convergence properties**
> Thanks for your comment! If we understand correctly, you are referring to convergence rate analysis. To our knowledge, this remains an open question, except for very simple shallow MLP architectures. Achieving a theoretical breakthrough in this area would significantly advance our understanding and provide valuable insights into the behavior of the proposed method. Investigating these convergence properties is an important aspect of our future research plans.
>
> ### **Q1. Why chose Gaussian functions as feature embedding? Would other embedding functions fail?**
>
> Gaussian functions possess several excellent properties, such as the fact that mixtures of Gaussians are universal approximators. Our choice was also partly inspired by the recent success of Gaussian functions as graphical primitives (e.g., 3D Gaussian Splatting), which have demonstrated remarkable success in representing intricate and high-frequency details. To provide further context, we have also tested other embedding functions as part of our investigation.
>
> - Multivariate Cauchy function :  $\left(\frac{\Gamma\left(\frac{1+k}{2} \right)}{\Gamma\left(\frac{1}{2} \right)\pi^{\frac{k}{2}} |\Sigma|^{\frac{1}{2}} [1 + (x-\mu)^{T}\Sigma^{-1}(x-\mu)]^{\frac{1+k}{2}}}\right)$, ($k=$dimension). We tested on the Klein-Gordon Eq. through the multivariate 3D Cauchy function and obtained an L2 Relative Error of 1.05e-02 (PIG: 2.76e-03).
>
>
> - Sine functions: As the reviewer 77oA also suggested, we provided the results with sinc functions in the table below. The more details are provided in the Appendix A10.
>
> |   | 2D Helmholtz  | 3D Flow-Mixing  | 3D Klein-Gordon |
> |---|---|---|---|
> | SIREN | 1.68e-03 $\pm$ 2.02e-03 | 1.22e-02 $\pm$ 4.17e-03 | 1.18e-01 $\pm$  4.88e-02 |
> | SIREN + MLP| 1.31e-03 $\pm$ 8.26e-04 | 2.80e-02 $\pm$ 2.50e-02 | 1.04e-01 $\pm$  8.61e-02 |
> | PIG + SIREN | 1.37e-05 $\pm$ 1.64e-06 | 1.28e-03 $\pm$ 1.09e-04 | 2.37e-02 $\pm$ 4.62e-03 |
> | PIG + tanh | 4.13e-05 $\pm$ 2.59e-05 | 4.51e-04 $\pm$ 1.74e-04 | 2.76e-03 $\pm$ 4.27e-04 |
>
> - SIREN: 4 hidden layers MLP w/ 256 units, sine activation functions
> - SIREN + MLP: similar setting as PIG, replacing Gaussian basis with sine basis, followed by tiny MLP
> - PIG + SIREN: Gaussian embedding + sine basis in the tiny MLP
> - PIG + tanh: Gaussian embedding + tanh basis in the tiny MLP

---

> > ### Comment · Reviewer_uWK5 · 2024-11-22
> >
> > Thank you very much for your reply. I will maintain my score.

---

> > > ### Author Response · Authors · 2024-11-25
> > >
> > > Thank you very much for your response.

---

### Official Review · Reviewer_vuTU · 2024-10-18

**Soundness:** 4
**Presentation:** 4
**Contribution:** 3
**Rating:** 8
**Confidence:** 4

**Summary:**

Physics-Informed Gaussians (PIG) is proposed, an efficient and accurate PDE solver that utilizes
learnable Gaussian feature embeddings and a lightweight neural network, counteracting the problems inherent in previous static parametric grid approaches. PIG model achieves competitive accuracy and faster convergence with fewer parameters compared to state-of-the-art methods.

**Strengths:**

- The paper is well-written and easy to follow. I enjoyed reading it.

- Motivation is clear.

- Comparisons with previous works are well-discussed.

- The proposed model is supported by both theoretical and empirical evidence.

- Error bars are provided, enhancing the reliability of the experimental results.

- Code is provided for reproducibility.

**Weaknesses:**

- (Minor) Figures should be vector images to avoid pixelization.

- While I found no significant weaknesses, there may be room to further expand the discussion on the novelty and its potential impact on future research.

**Questions:**

- (Line 245-) "Similar to the previous parametric grid methods, which obtain feature embeddings by interpolating only neighboring vertices, this locality encourages the model to capture high-frequency details by effectively alleviating spectral bias": Is it possible to mathematically prove that "this locality encourages the model to capture high-frequency details by effectively alleviating spectral bias" in some sense? This is just a question out of curiosity.

- Is the title of Section 3.2.2 relevant?

- Why is tanh used in the architecture (it is fine if there is no specific reason)? What about sin or swish activations?

- Is there any issues about mode collapses ($\mu_i \rightarrow \mu_j$ for all $i$ and $j$ and/or $\sigma_i \rightarrow 0$ or $\infty$)?

- I think the update scales of $W$ and $\theta$ differ significantly from those of $\mu_i$ and $\sigma_i$. The optimal updates for $\mu_i$ and $\sigma_i$ likely depend on the scale of the simulation. Did the authors encounter any issues related to this?

- It may be off-topic, but is the proposed method applicable to high-dimensional PDEs, such as $d\gtrsim100$? How significant is the computational cost?

---

> ### Author Response · Authors · 2024-11-22
> **W1, Q1**
>
> ### **W1. Figures should be vector images to avoid pixelization.**
> We will reflect the correction in the final version. Thank you for your feedback.
>
> ### **Q1. Is it possible to mathematically prove that "this locality encourages the model to capture high-frequency details by effectively alleviating spectral bias" in some sense?**
>
> Locality in the context of parametric grid methods means that the feature embeddings depend only on neighboring vertices. This encourages the model to focus on local variations, which often correspond to high-frequency components in the input. The Universal Approximation Theorem states that neural networks, with sufficient capacity, can approximate any function to an arbitrary degree of accuracy. However, the ability to efficiently represent high-frequency components often depends on the architecture and the specific properties of the network.
> In [Yarotsky, Neural Networks, 2017], the author showed that neural networks with certain activation functions (e.g., ReLU) and locality properties can capture high-frequency details when appropriately trained. The locality constraint ensures that the network learns functions that are sensitive to rapid changes in the input, thereby mitigating spectral bias. Spectral bias refers to the tendency of standard neural networks to prioritize learning lower-frequency components of a function during training. Locality helps address this by constraining the receptive field and focusing on finer-scale features.
> Building on this, we provide a conceptual demonstration of how locality enhances the approximation ability of neural networks by leveraging a partition of unity framework. This approach illustrates how locality facilitate the capture of details, including addressing high-frequency function. Furthermore, this concept may be extended to include local Gaussian approximations, broadening its applicability.
>
> Let $\ M \$ be a positive integer. To construct an approximation, we first define a partition of unity over the domain $\[0,1]^d\$ using a grid of $\ (M+1)^d \$ functions $\ \psi_q \$ such that $\ \sum_q \psi_q(\mathbf{z}) \equiv 1, \quad \forall \mathbf{z} \in [0,1]^d\$. Here, $\ q = (q_1, \ldots, q_d) \in$ {0, 1, $\ldots$, M\}$^d$ \, and the functions $\ \psi_q \$ are defined as $\ \psi_q(\mathbf{z}) = \prod_{i=1}^d \varphi\left(3M \left(z_i - \frac{q_i}{M}\right)\right), \$ where the function $\ \varphi(x) \$ is a local shape function given by $\ \varphi(x) $ = (1 (if |x| < 1) or 2 -  |x| (if 1 $\leq |x| \leq 2$) or 0 (if |x| > 2)). The functions $\ \psi_q \$ are compactly supported, satisfying $\mathrm{supp}$ $\psi_q$ $\subseteq$ \{ $\mathbf{z}$ : |$z_i$ - $\frac{q_i}{M}$| < $\frac{1}{M}$, $ \forall i $} and are uniformly bounded as $\|\psi_q\|_\infty = 1, \forall q$.
>
> Next, for any $\ q \in$ {0, $\ldots$, M\}$^d$ , we construct the degree-$\(p-1)\$ Taylor polynomial of a function $\ g \$ around the point $\mathbf{z} = \frac{\mathbf{q}}{M}$. The Taylor polynomial is given by
>
> $T_q(\mathbf{z}) = \sum_{|\mathbf{m}| < p} \frac{\partial^{\mathbf{m}} g}{\mathbf{m}!} \bigg|_{\mathbf{z} = \frac{\mathbf{q}}{M}} \left( \mathbf{z} - \frac{\mathbf{q}}{M} \right)^{\mathbf{m}},$
>
> where $\mathbf{m}! = \prod_{i=1}^d m_i! $ and $\left( \mathbf{z} - \frac{\mathbf{q}}{M} \right)^{\mathbf{m}} = \prod_{i=1}^d \left( z_i - \frac{q_i}{M} \right)^{m_i}$.
>
> Using this construction, an approximation to  $g$ is defined as $g_1(\mathbf{z}) = \sum_{q \in \(0, \ldots, M\)^d} \psi_q(\mathbf{z}) T_q(\mathbf{z}).$ To bound the error of this approximation, we note that $|g(\mathbf{z}) - g_1(\mathbf{z})| = \left| \sum_q \psi_q(\mathbf{z}) \big(g(\mathbf{z}) - T_q(\mathbf{z})\big) \right|$. Since each $\psi_q$ is supported locally, this sum reduces to $\sum_{\substack{q : |z_i - \frac{q_i}{M}| < \frac{1}{M}, \forall i}} |g(\mathbf{z}) - T_q(\mathbf{z})|$.
>
> Using the compact support of $ \psi_q $, this is further bounded by $2^d \max_{\substack{q : |z_i - \frac{q_i}{M}| < \frac{1}{M}, \forall i}} |g(\mathbf{z}) - T_q(\mathbf{z})|$. Applying the Taylor remainder theorem, we obtain
>
> $|g(\mathbf{z}) - T_q(\mathbf{z})| \leq \frac{d^p}{p!} \left( \frac{1}{M} \right)^p \max_{|\mathbf{m}| = p}  \text{ess} \sup_{\mathbf{z} \in [0,1]^d} |\partial^{\mathbf{m}} g(\mathbf{z})|$,
>
> and thus the total error is bounded as $|g(\mathbf{z}) - g_1(\mathbf{z})| \leq \frac{2^d d^p}{p!} \left( \frac{1}{M} \right)^p$.
>
> This result demonstrates that the approximation error decreases with increasing $M$ and $p$, with a convergence rate dictated by the resolution of the partition.

---

> ### Author Response · Authors · 2024-11-22
> **Q2, Q3, Q4, Q5, Q6**
>
> ### **Q2. Is the title of Section 3.2.2 relevant?**
> Thank you for your feedback. The topic "The approximation of PDE" is a covering expression of 3.2.1 and 3.2.2. We revised 3.2.2 to a more detailed expression, “Generating the Solution from Learnable Gaussians With Lightweight Neural Network”.
>
> ### **Q3. Why is tanh used in the architecture? What about sin or swish activations?**
> The choice of the tanh activation function is primarily motivated by its extensive use in various PINN architectures, and the baseline methods we compared have also utilized tanh. Hence, we followed this established approach for consistency and fair comparison. Regarding the use of sin, there is a comparative study of using sin and tanh (Random weight factorization, Wang et al., arXiv 2022). The results of the experiment with the sine activation function and swish activation function are as follows.
>
> |   | 2D Helmholtz  | 3D Klein-Gordon  | 3D Flow-Mixing  |
> |---|---|---|---|
> | Tanh  |  4.13e-05 $\pm$ 2.59e-05 |  2.76e-03  $\pm$ 4.27e-04 | 4.51e-04 $\pm$ 1.74e-04  |
> | Swish  |  5.21e-05 $\pm$ 2.91e-05 | 3.55e-03 $\pm$ 1.34e-03  | 4.36e-04 $\pm$ 3.82e-05  |
> | Sine  |  3.09e-05 $\pm$ 3.34e-06  | 6.08e-03 $\pm$ 3.15e-03  | 5.20e-04 $\pm$ 2.09e-05  |
> | SIREN ($\sin(3x)$)  |  1.37e-05 $\pm$ 1.64e-06  | 2.37e-02 $\pm$ 4.62e-03  | 1.28e-03 $\pm$ 1.09e-04  |
>
> ### **Q4. Is there any issues about mode collapses $\mu_{i} \rightarrow \mu_{j}$ for all i and j or $\sigma \rightarrow 0 \text{ or } \infty$?**
> Regarding the Gaussian overlap, while nothing prevents it from happening, we believe this would not pose significant issues. The weighting factors associated with each Gaussian determine their contributions to the solution, effectively resolving conflicts even when Gaussians are very close to each other. As for the variances, while we have not observed any serious issues in our experiments, numerical errors could potentially arise when the variances become extremely small.
>
> To further investigate, we analyzed the distances between Gaussians to assess overlap and examined whether the sigma values converge to zero. Details of this analysis are provided in Appendix Figure 15. The histogram results indicate that there are some Gaussians located very close to each other. The minimum sigma values observed were on the order of $10^{-6}$, suggesting that extreme cases are rare.
>
> ### **Q5. Unmatched gradient scales for $\theta$ and $\mu, \sigma$?**
>
> Thanks for your insightful comment. In our current experimental settings, the simulation domains for all PDEs are in similar scales, such as [0,1$]^d$, and our current hyperparameter settings appear to be well-suited for this input domain scale. We have tested with different scales, such as [0,10$]^d$, and observed meaningful differences in convergence behavior. While we believe that adaptive learning rate optimizers (adam) might partially address this issue, explicitly incorporating specialized techniques could potentially improve convergence behavior for varying simulation scales. We appreciate you bringing this up, and we plan to investigate further to develop a more advanced algorithm.
>
> ### **Q6. High-dimensional PDEs. (100 Dimensional PDEs)**
> Thanks for your suggestion! We tested PIG on 100D Allen-Cahn and 100D Poisson equations using the SDGD method (Hu et al. Tackling the Curse of Dimensionality with PINNs, Neural Networks, 2024). The following table presents the results, and the L2 error curve during training is provided in the Appendix (Figure 11) of the updated pdf file.
>
> |   | 100D Allen-Cahn | 100D Poisson |
> |---|---|---|
> | Error | 8.88e-03 | 8.42e-03|
> | # Gaussians | 3000 | 3000 |
> | Memory (GB) | 15.7 | 22.6|
> | Time (minutes) | 12 | 21|

---

> > ### Comment · Reviewer_vuTU · 2024-11-23
> >
> > Thank you for your thorough discussion and additional experiments. The additional theoretical discussion is relevant and should be included in the paper. I have also gone through other reviews, and the author's response answered my questions. I maintain my score because it is already kind of saturated. I support the acceptance of the paper.

---

> > > ### Author Response · Authors · 2024-11-25
> > >
> > > Thank you very much for your thoughtful comments and interest in our paper. We will include the theoretical discussion in the final version of the paper. Thank you very much!

---

### Official Review · Reviewer_SQ8v · 2024-11-03

**Soundness:** 3
**Presentation:** 4
**Contribution:** 2
**Rating:** 6
**Confidence:** 4

**Summary:**

The authors propose the Physics-Informed Gaussian (PIG) that learns feature embeddings of input coordinates, using a mixture of Gaussian functions. Compared to PINN, this paper applies Gaussians to input coordinates and makes the mean and covariance matrix learnable. The locality of Gaussians encourages the model to capture high-frequency details by effectively alleviating spectral bias.

**Strengths:**

The paper is well-written and easy to follow. The code and experiments are of good quality. It provides a more concise and more parameter efficient feature embedding method compared to previous parametric grid representations method.

**Weaknesses:**

- The primary concern lies in the originality of this method. Though it’s considered as a feature embedding, mathematically, it is in the same spirit of a KAN layer with RBF. Also, it is not uncommon to consider Gaussian embedding in ML.

- The paper states that the approach "dynamically adjusts the positions and shapes of the Gaussians during training" with parameters "iteratively updated throughout the training process." The claims in the paper give an impression of adaptive sampling method based on the training. However, the approach does not actually take into account the residual or the loss function landscape. It would be better to provide a more rigid description on this. Also, in the experiments, the method is compared with different PINN approaches for each PDE, and the authors report the results without conducting experiments by themselves again. Though “the sensitivity of PINN variants to hyperparameters complicates fair comparisons”, it’s still necessary to conduct experiments with same setup and number of parameters to ensure a fair comparison. The average error of PIG is generally higher, but the authors only highlight the best of PIG. PIG is claimed to “enable more accurate and efficient approximations”, but the efficiency is not clear. Hence, it seems a lot of conclusions in this paper are overstated.

- I would like to see how the spectral bias can be alleviated, either through experimental evidence or analytical justification.

- For the 2D and 3D cases, what is the mathematical formulation for the Gaussian embedding? How is the parameter being initialized? Additionally, how are the boundary conditions (BC) being constrained?

- The paper mentions that "computational costs per iteration of our method are significantly lower than JAX-PI." A detailed report of these computational costs would be beneficial.

**Questions:**

- I would be interested to see how the authors address the weaknesses above. Could authors provide additional evidence or context to support the novelty?
- Have the authors conducted any experiments to demonstrate how the method alleviates spectral bias?

---

> ### Author Response · Authors · 2024-11-22
> **W1, W2**
>
> ### **W1. The primary concern lies in the originality of this method. Though it’s considered a feature embedding, mathematically, it is in the same spirit as a KAN layer with RBF.**
>
> - Thank you for your insightful comments. Regarding the relationship between the KAN layer and RBF, we acknowledge the similarity in spirit between the two approaches. However, there are notable differences. A width $n$ KAN layer can be represented as follows:
>
> $$
> \Phi(x_1, \dots, x_d)
> :=
> [\sum_{i=1}^d \phi_{1, i}(x_i) \cdots \sum_{i=1}^d \phi_{n, i}(x_i)]^T
> $$
>
> In contrast, our Gaussian feature embedding takes the form of
>
> $$
> [N\left(x_1, \dots, x_d; \ \mu_1, \Sigma_1\right) \cdots N\left(x_1, \dots, x_d; \ \mu_n, \Sigma_n\right)]^T
> $$
>
> However, the function $(x_1, \dots, x_d) \mapsto N\left(x_1, \dots, x_d; \ \mu, \Sigma\right)$ is not a sum of 1D functions.
>
> We also agree that Gaussian feature embedding is not entirely new, as the history of the RBF networks dates back to the 1980's. (Lowe and Broomhead, Multivariable functional interpolation and adaptive networks, Complex Systems, 1988.) Yet our work tries to empirically validate the effectiveness of Gaussian embedding in the context of PINNs. We would also like to emphasize that we present PIG as a more flexible, efficient, and accurate parametric grid for PDE solvers. Moreover, we revisit the expressive power of the mixture of Gaussians, drawing inspiration from recent breakthroughs in 3D computer graphics, demonstrating the promising performance on various challenging PDEs.
>
> ### **W2. Impression of an adaptive sampling method, conducting experiments with the same setup and number of parameters**
> - To clarify, during the training process, we update the $\mu$ and $\Sigma$ parameters of the Gaussians. This means that the Gaussians adjust their positions and shapes during training. Since we use gradient descent, we update  $\mu$ and $\Sigma$ to minimize the overall loss, which includes the PDE residual loss.
> To further account for the loss surface, we can adopt higher-order gradient descent algorithms, such as L-BFGS. However, while these methods generally offer more precise updates, they tend to incur higher computational costs per iteration. In our experiments, we did not observe significant performance improvements with L-BFGS. Nevertheless, exploring more advanced optimization techniques remains an exciting research direction.
> If you are referring to 'adaptive sampling of collocation points,' we would like to clarify that we do not employ such techniques. Currently, collocation points are randomly sampled during several iterations. Investigating how PIG performs under adaptive collocation point sampling would indeed be an interesting avenue for future research. Thanks for your valuable suggestion!
>
> - The reason we did not provide in-house training results for other methods is to avoid potential criticism for not making sufficient efforts to optimize the baseline methods. Instead, we report the results directly from the original papers to ensure fairness. This leads to comparisons against different PINN methods for different PDEs (if a baseline method did not report the result for a particular PDE, we chose to compare to other baselines).
>
> - We report the mean error and standard deviation of our results, whereas many other works typically report only the best error. We believe this provides a more comprehensive and realistic evaluation, which we consider a strength rather than a limitation.
>
> - We conducted baseline experiments using a similar number of parameters as the PIG model. In the order of the table below, PIG used [2897, 112097, 112097, 16096, 20097] parameters whereas the baseline models used [3066, 114486, 118824, 16384, 20268] parameters. The results demonstrate that PIG performs significantly better than baseline methods for all equations. The baseline model for 3D Klein-Gordon, 3D Flow-Mixing, and 3D Nonlinear-Diffusion is SPINN, the baseline for 2D Helmholtz is PIXEL, and the baseline for 2D Allen-Cahn is JAXPI.
>
> |   | 3D Klein-Gordon  | 3D Flow-Mixing  | 3D Nonlinear-Diffusion | 2D Helmholtz | 2D Allen-Cahn |
> |---|---|---|---|---|---|
> | Baselines | 1.15e-02 $\pm$ 4.09e-03 | 8.62e-02 $\pm$ 3.36e-03 | 3.42e-02 $\pm$ 2.22e-03 | 5.39e-03 $\pm$ 2.15e-03 | 1.80e-04 $\pm$ 5.96e-05 |
> | PIG | 2.76e-03 $\pm$ 4.27e-04 | 4.51e-04 $\pm$ 1.74e-04 | 2.69e-03 $\pm$ 6.55e-04 | 4.13e-05 $\pm$ 2.59e-05 | 1.04e-04 $\pm$ 4.12e-05 |

---

> ### Author Response · Authors · 2024-11-22
> **W3 & Q2, W4, W5**
>
> ### **W3. Q2 How the spectral bias can be alleviated?.**
> To address spectral bias, we conducted experiments on the high-frequency Helmholtz equation $(a_1 = 10, a_2 = 10)$. Our method achieved an $L^2$ relative error of $7.09 \times 10^{-3}$. Notably, this equation cannot be learned at all using standard PINNs. The results are presented in Appendix Figure 14.
>
> ### **W4, For the 2D and 3D cases, what is the mathematical formulation for the Gaussian embedding? How is the parameter being initialized? Additionally, how are the boundary conditions (BC) being constrained?**
>
> Thank you for seeking clarification on this point. The mathematical formulation of our Gaussian embedding
>
> $\texttt{FE}_\phi: \mathbb{R}^d \rightarrow \mathbb{R}^k$ becomes a length $k$ vector whose $j$-th element is $\sum\_{i=1}^N f\_i^j \exp\left( (x-\mu_i^j)^T (\Sigma_i^j)^{-1} (x - \mu_i^j) \right)$.
>
> - For 2D, $\mu_i^j \in \mathbb{R}^{2}$,  $\Sigma_i^j \in \mathbb{S}^{2}_{++}$,
> - For 3D, $\mu_i^j \in \mathbb{R}^{3}$,  $\Sigma_i^j \in \mathbb{S}^{3}_{++}$.
> - Further mathematical details can be found in Equation (4) of the paper, where the case of \(k = 1\) is considered.
>
> The parameter initialization process was carried out empirically. For the mean values $(\mu)$, we selected values of 1 or 2, while for the $\Sigma$ values (sigma), we chose values within the range of $[0.01, 1]$. More details regarding this initialization process are provided in Appendix A of the paper.
>
> For the boundary conditions, we primarily employed Dirichlet boundary conditions across most of our experiments. To enforce these constraints, we created boundary point data and incorporated it into the loss function, effectively performing regression on the boundary data to satisfy the conditions. This approach ensured that the solutions adhered to the prescribed boundary values throughout training. For the Allen-Cahn equation, however, we achieved excellent results without explicitly imposing constraints on the boundary. In this case, we relied solely on the residual loss and initial loss, demonstrating that the model was able to implicitly satisfy the boundary behavior through its learning process.
>
> ### **W5. The paper mentions that "computational costs per iteration of our method are significantly lower than JAX-PI." A detailed report of these computational costs would be beneficial.**
>
> We appreciate your interest in the computational cost analysis. For example, in the case of the Allen-Cahn equation, our method (PIG) demonstrates significant advantages over JAX-PI in terms of both computational speed and memory efficiency. Specifically, JAX-PI requires 0.01666 seconds per iteration, which is **2.30 times slower** compared to PIG's 0.0074 seconds per iteration. Additionally, JAX-PI consumes 22.9GB of memory, which is **11.68 times larger** than the 1.96GB required by PIG. These results clearly illustrate the efficiency of PIG, making it a much more practical and scalable solution for complex problems such as the Allen-Cahn equation.

---

### Official Review · Reviewer_77oA · 2024-11-04

**Soundness:** 3
**Presentation:** 3
**Contribution:** 3
**Rating:** 6
**Confidence:** 4

**Summary:**

In this paper, the authors use deep Gaussian mixture models as parametric mesh representations to replace MLP in PINN, so that it performs better for high-frequency and non-linear components. Experiments for PDEs such as Allen-Cahn, Helmholtz, Nonlinear diffusion, etc., are conducted compared with several previous PINN derivatives, showing that the proposed method achieves sota solution accuracy.

**Strengths:**

- A theoretical proof of the universal approximation capability for the proposed method is provided.
- The experiments are diverse, the baselines are sufficient, and the ablation study is detailed.
- The writing is overall clear and detailed.

**Weaknesses:**

- Though diverse and rich, the experimental settings are all relatively simple, and lack comparisons to traditional methods.
- Correct me if I am wrong, but I think only the solution accuracy is provided, without the computation time?

**Questions:**

- With the work SIREN in mind, I wonder if the Gaussian basis is the best choice?
- The idea is quite similar to 3D Gaussian splatting in rendering. However, there are differences. One easy way to explain is that, we can think of the volume rendering equation as a specific type of governing physics equation. Then the success of applying parametric mixed Gaussian in one specific type of equations doesn't mean it is the best idea for others. For example, there can be long-range interactions for certain PDEs, while Gaussian functions are local and lack such representation capability. I would like to have the authors' opinions on such comments.

---

> ### Author Response · Authors · 2024-11-22
>
> ### **W1. The experimental settings are all relatively simple and lack comparisons to traditional methods**
> To provide further information, we have performed various experiments in more challenging setups.
>
> - First, we tested PIG on a non-simple Lid-Driven Cavity equation, a well-known challenging problem due to its singular behavior. We obtained superior performance (4.04e-04) than the Parametric Grid baseline (1.22e-03). The experimental results are provided in the Appendix (Figure 12, 13) of the updated submission pdf file.
> - Second, we solved the 100-dimensional Poisson equation and Allen-Cahn equation using PIG combined with the SDGD method (proposed by Hu et al. Tackling the Curse of Dimensionality with PINNs, Neural Networks, 2024.).The 100D Allen-Cahn achieved 8.88e-03 L2 relative error and 100D Poisson achieved 8.42e-03 L2 relative error. The L2 relative error curve during training is presented in the Appendix (Figure 11).
> - Third, we also showed that PIG can solve an inverse problem of finding the unknown coefficient values ​​(\lambda = 5) of the Allen-Cahn equation, and the PIG converges faster than PINNs (Figure 10 in the Appendix).
>
> In terms of comparison to traditional methods (which we understand you refer to numerical methods), we acknowledge that the accuracy of PIG or other PINN variants has not yet reached the level of traditional numerical methods in many cases. However, we believe that the field is in an exploratory phase, and significant progress is being made to address these limitations.
>
> One of the key strengths of PIG or PINNs lies in their versatility and unique properties compared to traditional methods. For example, PINNs excel in handling high-dimensional problems, whereas traditional methods often face severe scalability issues. Additionally, PIG or PINNs provide a unified optimization framework that seamlessly addresses both forward and inverse problems. We have provided experimental evidence that PIG can effectively handle both cases (high-dim and inverse problems), and we hope we address your concerns and highlight the strengths of the proposed PIG.
>
> ### **W2. Computation time**
> The table below presents computation time per iteration and memory usage. Please also note that JAX-PI requires 0.01666 seconds (**x2.30 slower**) per iteration and 22.9GB (**x11.68 larger**) of memory for the Allen-Cahn equation.
>
> |   | 3D Klein-Gordon | 2D Allen-Cahn  | Nonlinear-Diffusion  | 2D Helmholtz | 3D Flow-Mixing |
> |---|---|---|---|---|---|
> | Time per iteration (s) | 0.0025 | 0.0074 | 0.042 | 0.050 | 0.12 |
> | Memory (GB) | 0.7 | 1.96 | 4.9 | 3 | 17.8 |
>
> ### **Q1. Is Gaussian basis the best choice? (Comparison with SIREN)**
> We acknowledge that the proposed Gaussian basis approach cannot be definitively claimed as the best choice. We have provided both empirical evidence demonstrating its effectiveness and a theoretical analysis supporting its properties. Thank you for pointing out SIREN. Based on your comment, we conducted additional experiments using sine basis functions. The results of these experiments are presented in the table below, and more details are provided in the Appendix A10.
>
> - SIREN: 4 layer MLP w/ 256 units, sine activation functions
> - SIREN + MLP: similar setting as PIG, replacing Gaussian basis with sine basis, followed by tiny MLP
> - PIG + SIREN: Gaussian embedding + sine basis in the tiny MLP
> - PIG + tanh: Gaussian embedding + tanh basis in the tiny MLP
>
> |   | 2D Helmholtz  | 3D Flow-Mixing  | 3D Klein-Gordon |
> |---|---|---|---|
> | SIREN | 1.68e-03 $\pm$ 2.02e-03 | 1.22e-02 $\pm$ 4.17e-03 | 1.18e-01 $\pm$  4.88e-02 |
> | SIREN + MLP| 1.31e-03 $\pm$ 8.26e-04 | 2.80e-02 $\pm$ 2.50e-02 | 1.04e-01 $\pm$  8.61e-02 |
> | PIG + SIREN | 1.37e-05 $\pm$ 1.64e-06 | 1.28e-03 $\pm$ 1.09e-04 | 2.37e-02 $\pm$ 4.62e-03 |
> | PIG + tanh | 4.13e-05 $\pm$ 2.59e-05 | 4.51e-04 $\pm$ 1.74e-04 | 2.76e-03 $\pm$ 4.27e-04 |
>
> ### **Q2. Volume rendering equation, Long-range interactions for PDEs**
> Thank you for raising this insightful question. We appreciate your thoughtful comments and perspective. Indeed, volumetric rendering can be interpreted as a governing equation, though, strictly speaking, it does not involve partial derivatives and may not be classified as a PDE in the traditional sense.
>
> In this work, we have focused on demonstrating that the proposed PIG framework performs effectively across a range of PDEs, showcasing its generalizability and applicability beyond specific types of governing equations. Regarding the local property of Gaussians, we would like to note that while Gaussian functions are often considered local, they can also exhibit global characteristics, enabling them to capture long-range interactions. In fact, in our experiments, we observed instances of very large Gaussians that span the entire time domain, effectively representing global interactions.

---

> ### Comment · Reviewer_77oA · 2024-11-25
> **Another quick question**
>
> Thank you for your hard work and detailed reply!
>
> Can I ask another quick question before fully going through all the discussions? I found a master thesis which is not cited in the paper. I wonder if the authors have read about it before and if you can highlight the difference and novelty:
>
> Rensen, Max, Michael Weinmann, Benno Buschmann, and Elmar Eisemann. "Physics-Informed Gaussian Splatting." (2024).

---

> > ### Author Response · Authors · 2024-11-27
> >
> > Thank you for bringing this reference to our attention; we sincerely appreciate it. Although we are unsure of its exact publication date, the paper states that it was published in July of this year, and we believe this is a concurrent work. While both approaches utilize Gaussians, the methods are fundamentally different in their algorithm, network architecture, and overall framework.
> >
> > First, their approach represents time-dependent PDEs by moving Gaussian positions and adjusting parameters starting from initial Gaussians. In contrast, our method proposes spatio-temporal multivariate Gaussians. Specifically, for (1+1)D PDEs, we employ two-dimensional Gaussians encompassing both spatial and temporal dimensions, whereas their method uses one-dimensional Gaussians.
> >
> > Second, the architecture of their work is primarily inspired by PointNet [2], consisting of encoder, aggregation, and decoder components. Their model adopts an autoregressive approach, taking the current Gaussian parameters to predict the delta of Gaussian parameters for the next time step. On the other hand, our proposed PIG framework utilizes Gaussians as irregular parametric grids and incorporates a tiny MLP to predict the solution directly. We believe that our method is more computationally efficient due to this design choice.
> >
> > Third, we compared ours to the referenced paper. We tested our PIG on a few PDEs presented in their paper and followed their experimental settings for fair comparison. The experimental results show that our model outperformed by a large margin, achieving significantly lower L2 relative errors of ***7.68e-04*** for the (2+1)D Single Gaussian Burgers Solution, ***1.08e-03*** for the (2+1)D Double Gaussian Burgers Solution, and ***9.04e-03*** for the (2+1)D Diffusion. For two Burgers equations, we reported the L2 errors from their paper. We have run their codes for the (2+1)D Diffusion equation since they did not report the L2 errors in the paper. The results can be found in Table 8, Figures 16, 17, and 18 of our updated paper.
> >
> > | | (2+1)D Burgers (Single Gaussian) | (2+1)D Burgers (Double Gaussian) | (2+1)D Diffusion |
> > |:---:|:---:|:---:|:---:|
> > | [1] L2 Rel. Error | 1.62e-01| 2.61e-01 | 3.97e+00 |
> > | Ours L2 Rel. Error | ***7.68e-04*** | ***1.08e-03*** | ***9.04e-03***|
> > | [1] Time (s) per Iter. | 1.50 | 1.68 | 4.20 |
> > | Ours Time (s) per Iter. | ***0.28*** | ***0.29*** | ***0.15*** |
> >
> > We will discuss this concurrent work in the revised manuscript to clarify the distinctions between the two approaches. If you have any further questions or require additional clarification, please do not hesitate to let us know. We would be delighted to address your concerns promptly.
> >
> > [1] Rensen, Max, Michael Weinmann, Benno Buschmann, and Elmar Eisemann. "Physics-Informed Gaussian Splatting." (2024).
> >
> > [2] Charles R. Qi et al. “PointNet: Deep Learning on Point Sets for 3D Classification and Segmentation”, Proceedings of the IEEE conference on computer vision and pattern recognition, (2017)

---

> ### Comment · Reviewer_77oA · 2024-11-30
>
> Thanks for your detailed reply, especially the extra one. All my questions are well addressed. I will maintain my already positive score.

---

> > ### Author Response · Authors · 2024-12-01
> >
> > Thank you very much for your response.

---

### Author Response · Authors · 2024-11-22

Thank you to all the reviewers for their reviews and constructive feedback. Before addressing each review individually, we will clarify some commonly raised points in this general response. All revised parts in the revised paper have been marked in red.

1. We anticipated potential questions regarding experiments involving full covariance cases, and as a result, we promptly initiated these experiments. Through these efforts, we have successfully enhanced the performance of the Full Covariance model. The final results of these experiments are presented in the table below.

|   | 2D Helmholtz | 3D Klein-Gordon  | 3D Nonlinear-Diffusion | 3D Flow-Mixing |
|---|---|---|---|---|
| Diagonal  | 1.10e-04 | 2.24e-03 | 4.13e-03 | 8.23e-04 |
| Full | 5.17e-05 | 1.81e-03 | 3.86e-03 | 3.48e-04 |
| # Gaussians  | 500 | 100 | 50 | 400 |


2. We conducted additional experiments on the 100-dimensional Allen-Cahn and Poisson equations. For the 100-dimensional Allen-Cahn equation, PIG achieved an $L^2$ relative error of $8.88 \times 10^{-3}$. Similarly, for the 100-dimensional Poisson equation, PIG achieved an $L^2$ relative error of $8.42 \times 10^{-3}$. The $L^2$ relative error curves for these experiments are provided in Appendix Figure 11.

3. We tested the Lid-Driven Cavity equation, a well-known challenging problem due to its singular behavior. The results demonstrated that our method (PIG) achieved superior $L^2$ relative error performance compared to the baseline model PGCAN using the Parametric Grid method (PIG: $4.04 \times 10^{-4}$, PGCAN: $1.22 \times 10^{-3}\)$. The detailed results and figures can be found in Appendix Figures 12 and 13.


4. We performed additional experiments on the inverse problem of the Allen-Cahn equation. These experiments showed that PIG converged significantly faster, as illustrated in Appendix Figure 10.

5. To address spectral bias, we conducted experiments on the high-frequency Helmholtz equation $(a_1 = 10, a_2 = 10\)$. Our method achieved an $L^2$ relative error of $7.09\times 10^{-3}$. Notably, this equation cannot be learned at all using standard PINNs. The results are presented in Appendix Figure 14.

7. Finally, we compared the computational time and memory usage of our method. For the Allen-Cahn equation, JAX-PI required 0.01666 seconds per iteration, which is **$2.30\times$** slower compared to the baseline, and consumed 22.9GB of memory, which is **$11.68\times$** larger. These details are summarized in the table below for clarity.

|   | 3D Klein-Gordon | 2D Allen-Cahn  | Nonlinear-Diffusion  | 2D Helmholtz | 3D Flow-Mixing |
|---|---|---|---|---|---|
| Time per iteration (s) | 0.0025 | 0.0074 | 0.042 | 0.050 | 0.12 |
| Memory (GB) | 0.7 | 1.96 | 4.9 | 3 | 17.8 |

---

### Meta-Review · Area_Chair_5mZj · 2024-12-24

**Metareview:**

This paper proposes a new version of PINNs to solve PDEs. The proposed method utilizes the Gaussian feature embeddings with a lightweight neural networks. A universal approximation theory is provided. Numerical experiments show the proposed method is effective compared with other standard methods.

This paper is well written. The theoretical analysis justifies plausibleness of the proposed method. The numerical experiments are comprehensive and well highlight effectiveness of the proposed method in comparison with the other existing methods. A detailed ablation study is given.

Thus, I recommend acceptance for this paper.

**Additional Comments On Reviewer Discussion:**

Some technical questions were raised by the reviewers. But, the authors addressed almost all those questions by giving a detailed explanation and showing additional numerical experiment results. Then, there does not remain any specific question.

---

### Decision · Program_Chairs · 2025-01-22

Accept (Poster)